# Activation of mitochondrial TUFM ameliorates metabolic dysregulation through coordinating autophagy induction

Dasol Kim [1,3], Hui-Yun Hwang [1,3], Eun Sun Ji[2], Jin Young Kim[2], Jong Shin Yoo[2] & Ho Jeong Kwon [1✉]

Disorders of autophagy, a key regulator of cellular homeostasis, cause a number of human diseases. Due to the role of autophagy in metabolic dysregulation, there is a need to identify autophagy regulators as therapeutic targets. To address this need, we conducted an autophagy phenotype-based screen and identified the natural compound kaempferide (Kaem) as an autophagy enhancer. Kaem promoted autophagy through translocation of transcription factor EB (TFEB) without MTOR perturbation, suggesting it is safe for administration. Moreover, Kaem accelerated lipid droplet degradation in a lysosomal activity-dependent manner in vitro and ameliorated metabolic dysregulation in a diet-induced obesity mouse model. To elucidate the mechanism underlying Kaem's biological activity, the target protein was identified via combined drug affinity responsive target stability and LC–MS/MS analyses. Kaem directly interacted with the mitochondrial elongation factor TUFM, and TUFM absence reversed Kaem-induced autophagy and lipid degradation. Kaem also induced mitochondrial reactive oxygen species (mtROS) to sequentially promote lysosomal $Ca^{2+}$ efflux, TFEB translocation and autophagy induction, suggesting a role of TUFM in mtROS regulation. Collectively, these results demonstrate that Kaem is a potential therapeutic candidate/chemical tool for treating metabolic dysregulation and reveal a role for TUFM in autophagy for metabolic regulation with lipid overload.

[1] Chemical Genomics Global Research Laboratory, Department of Biotechnology, College of Life Science and Biotechnology, Yonsei University, Seoul 03722, Republic of Korea. [2] Biomedical Omics Group, Korea Basic Science Institute, Ochang, Chungbuk 28119, Republic of Korea. [3] These authors contributed equally: Dasol Kim, Hui-Yun Hwang. ✉email: kwonhj@yonsei.ac.kr

Macroautophagy (hereafter "autophagy") is a conserved catabolic process that maintains cellular homeostasis via lysosomal hydrolysis. In autophagy, cells degrade abnormal organelles or invading extracellular pathogens to cope with stress and sustain cellular health. Thus, the physiologic roles of autophagy critically impact the maintenance of organismal metabolic homeostasis under pathophysiologic conditions[1]. Accordingly, several studies have reported that dysregulation of autophagy has a role in the development of metabolic disorders, suggesting that autophagy also has a crucial role in regulating whole-body metabolism[2–4].

Lipid droplets (LDs) have a pivotal role in maintaining energy flux between storage and catabolism in vivo. Abnormal intracellular over-loading with LDs in a diverse range of organs such as adipose tissue, liver, pancreas, and brain tends to cause metabolic dysfunction[5–7]. Therefore, balancing of lipid metabolism is critical, as it affects physiology systemically. As substrates of autophagy, LDs are degraded by lysosomal hydrolases. The process of lipolysis associated with autophagy is currently described as "lipophagy"[8]. Numerous studies have suggested targeting autophagy genetically or pharmacologically as a promising strategy for treating metabolic dysregulation[9–11]. For example, rapamycin, which inhibits the autophagy core regulatory factor MTOR, improves insulin resistance[12], and rescues the aged phenotype[13] in obese mice by activating autophagy. However, chronic administration of rapamycin reportedly causes metabolic impairment due to perturbation of the fundamental regulators of cell survival, MTORC1, and MTORC2[14].

The small molecule kaempferide (Kaem) is a natural compound of the flavonoid family. Kaem was identified in the present study as a hit by autophagy phenotype-based screening. Several previous studies reported a role for Kaem in biological fitness by protecting cells against various environmental stresses. However, little research has examined the role of Kaem in autophagy or metabolic dysregulation. Although a recent report suggested that extracts of *Chromolaena odorata* leaves, which contain Kaem, reduce lipid accumulation in 3T3-L1 adipocytes, whether the effect is mediated by Kaem remains unclear, as the underlying mechanism was not determined[15]. Here, we report that Kaem has a role in regulating metabolic fitness by enhancing autophagy. This effect does not involve perturbation of the major autophagy regulatory factor, MTOR, which has a crucial role in cellular growth, suggesting that MTOR-related side effects can be avoided. Although some of the protective effects of Kaem may be explained by its flavonoid-related anti-oxidant properties or by virtual screening of its binding partners[16,17], the mechanism underlying the biophysical interactions and biological activity of Kaem are largely unknown.

While several agents have been identified as autophagy modulators, identification of their molecular targets as a means of revealing the mechanism underlying their therapeutic effects has proven challenging. The conventional approach for target identification employs the use of affinity-based probes[18], which involves several limitations due to the immobilization procedure. The challenges associated with preparing modified chemicals engendered the development of alternative methods in which unmodified compounds are used such as drug affinity response target stability (DARTS), which exploits changes in the protease susceptibility of target proteins upon chemical binding[19–21]. In this study, we leveraged DARTS analysis with an LC–MS/MS quantitative proteomics approach as a label-free method to identify target proteins of the autophagy-enhancing natural compound Kaem. Using this approach, we identified the mitochondrial translation factor TUFM as a target of Kaem, and the physical interaction was then confirmed in vitro. TUFM induced autophagosome formation following Kaem treatment, thus demonstrating the biological relevance of TUFM in autophagy. In addition, LC–MS/MS analysis upon Kaem treatment provides insight on TUFM in mitochondrial reactive oxygen species (mtROS) regulation and a mechanism underlying Kaem-induced autophagy.

## Results

**Kaem activates autophagic degradation.** To identify agents for improving metabolic conditions via enhanced autophagy, we first conducted a screen for autophagy flux enhancers. HeLa cells were treated for 24 h with components of a chemical library comprising 658-natural compounds and then stained with acridine orange (AO) as an indicator of acidic lysosomes[22–25]. Discovering small molecules that enhance lysosomal functionality can be an effective strategy for targeting metabolic disorders such as obesity and diabetes, as they act as activators of autophagic-turnover[10,26]. AO staining is a well-known assay to examine the function and integrity of lysosome, which is also used to evaluate the status of autophagic flux[27,28] Correlation of the readout with AO fluorescence was validated via measuring the intensity of cells that are stained with different concentration of the staining (Supplementary Fig. 1a), and via checking that a positive control indatraline, which enhances lysosomal acidity[29], increased AO intensity 1.2-fold, and a negative control bafilomycin A1, which inhibits acidic lysosome by perturbing proton channel[24], decreased AO intensity 0.7-fold (Supplementary Fig. 1b–d). From this screen, 13 hit candidates were primarily selected, which increased AO intensity over 1.2 folds; kaempferide (Kaem), tiliroside, asiaticoside, arbutin, asperosaponin VI, astilbin, astragalin, aucubin, aurantio-obtusin, madecassoside, regaudioside A, resveratrol, and salidroside. To identify autophagy enhancers, previously reported autophagy regulators are excluded. Kaem was selected as a final hit compound among 5 candidates, based on covering the constraints of Lipinski's five rules (molecular weight: 300.06; XLogP: 2.2; no. of H-A: 6; H-D: 3; rotational bonds: 2) and further evaluations of autophagy activity (Fig. 1a and Supplementary Figs. 1b, 2a, b).

Kaem increased lysosomal acidity more than 1.4-fold compared with the DMSO vehicle control in the phenotypic screen (Supplementary Fig. 1b). Induction of lysosomal activity by Kaem was validated using AO staining by confocal microscopy, which indicated increased acidic vacuoles compared with the vehicle control, as determined by puncta counting quantification (Fig. 1b, c). For further evaluation of lysosomal active state to lead autophagy by Kaem, other fluorescent probes such as BODIPY FL-pepstatin A[30,31], double quenched BSA (DQ-BSA)[32], and lysotracker were investigated. BODIPY FL-pepstatin A labeling within active cathepsin-positive vacuoles was markedly increased by rapamycin and Kaem treatment whereas it was almost completely abolished by bafilomycin A1 treatment (Supplementary Fig. 3a). In DQ-BSA assess, Kaem-treated cells indicated increased number and intensity of fluorescent vacuoles where lysosomal proteolysis enhanced turnover of autophagy-cargo (Supplementary Fig. 3b). In the lysotracker analysis, Kaem increased lysosomal acidic puncta whereas lysosomotrophic agent, $NH_4Cl$, diminished all acidic vacuoles (Supplementary Fig. 3c). Kaem also induced the conversion of LC3 (microtubule-associated light chain protein type 3) from the cytosolic I form to the vesicular II form and reduced the level of the autophagy substrate protein SQSTM1/p62, which indicated degradation by autophagy (Fig. 1d–f and Supplementary Fig. 3d). As the clearance of autophagosomes is pivotal for the completion of autophagy[33], we further investigated the autophagy turnover activity of Kaem. LC3-II generated via Kaem-mediated conversion accumulated upon co-treatment with the lysosomotropic

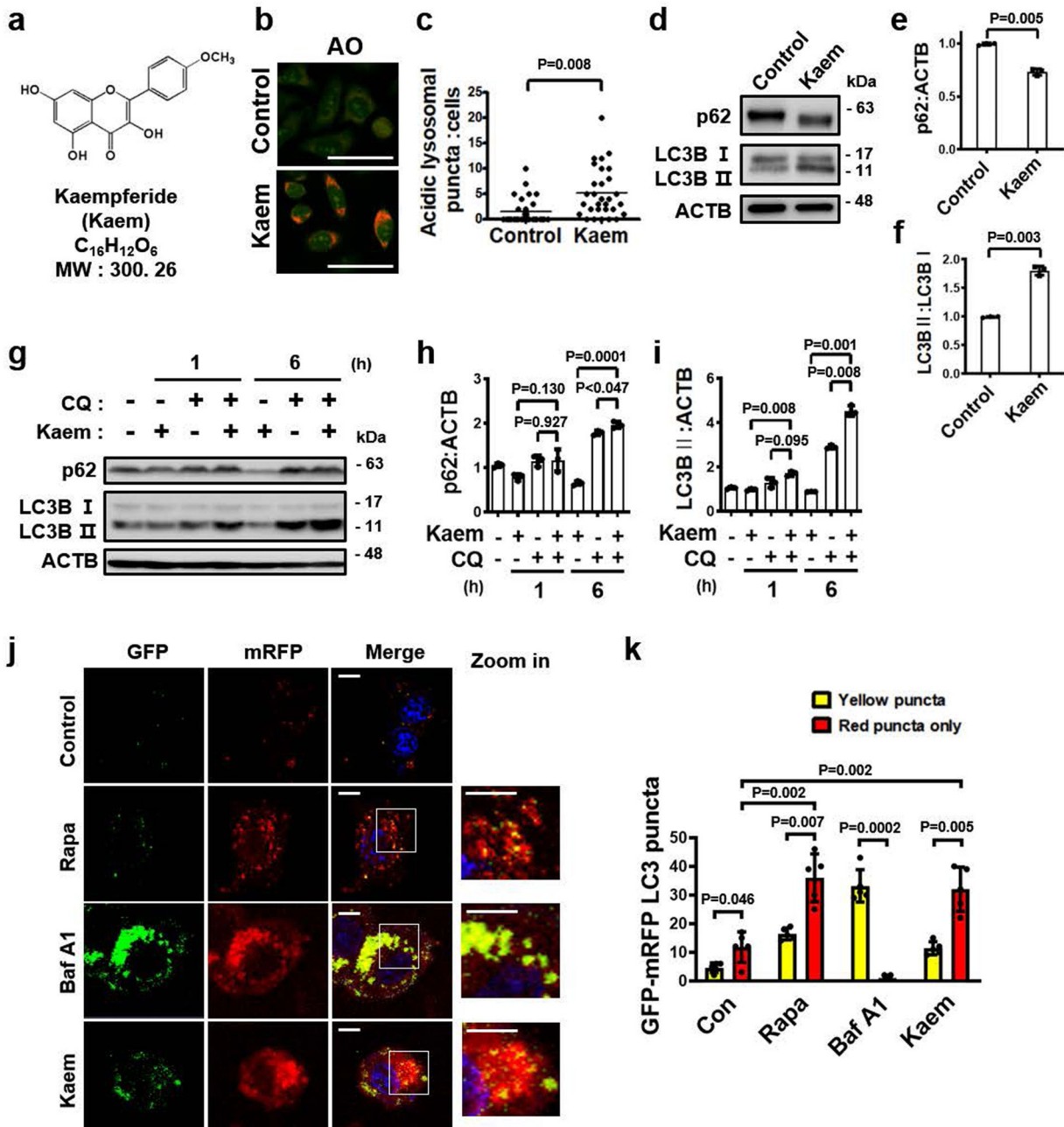

**Fig. 1 Kaempferide (Kaem) identified as a hit compound based on lysosomal activity promotes autophagy flux in HeLa cells. a** Chemical structure of the autophagy inducer Kaem. **b**, **c** DMSO control or Kaem-treated HeLa cells were stained with acridine orange (AO), and confocal microscopy was performed (**b**). Graph shows mean ± SD (n = 30) of acidic vesicles per cell (**c**). Kaem, 20 μM. Scale bar, 50 μm. **d–f** Hela cells treated with DMSO control or Kaem for 24 h. Cell extract was subjected to western blot analysis using antibodies against LC3B and p62. Representative images (**d**) and intensity of p62 (**e**) and LC3B (**f**) immunoblot bands normalized to ACTB. Kaem, 20 μM. Graph shows mean ± SD from three independent experiments. **g–i** Kaem treatment in the presence/absence of chloroquine (CQ). Cell extract was subjected to western blot analysis using antibodies against LC3B and p62. Representative images (**g**) and intensity of p62 (**h**) and LC3B (**i**) immunoblot bands normalized to ACTB. Kaem, 20 μM; CQ, 10 μM. Graph shows mean ± SD from three independent experiments. **j**, **k** HeLa cells transfected with mRFP-GFP-LC3 were treated with DMSO control, rapamycin (Rapa), bafilomycin A1 (Baf A1), and Kaem for 48 h, respectively, and confocal microscopy was performed. Representative images (**j**) and number of yellow (autophagosome) and red puncta (autolysosome) (**k**). Kaem, 20 μM; Rapa, 10 μM; BafA1, 10 nM. Scale bar, 10 μm. Graph shows mean ± SD (n = 5). Statistical significance was assessed by Student's *t*-test. ***P < 0.001; **P < 0.01; *P < 0.05.

agent chloroquine (CQ), whereas reduction of p62 protein by Kaem was abolished upon CQ treatment, indicating that co-treatment of cells with CQ and Kaem further increases the levels of these proteins compared to treatment with Kaem or CQ alone (Fig. 1g–i). This result indicated that Kaem promotes autophagosomal clearance in a lysosome-dependent manner. In cells expressing GFP-mRFP LC3, Kaem increased the mRFP/GFP puncta ratio. Kaem promoted the formation of autolysosomes (red fluorescence), which also indicates the induction of autophagy flux (Fig. 1j, k).

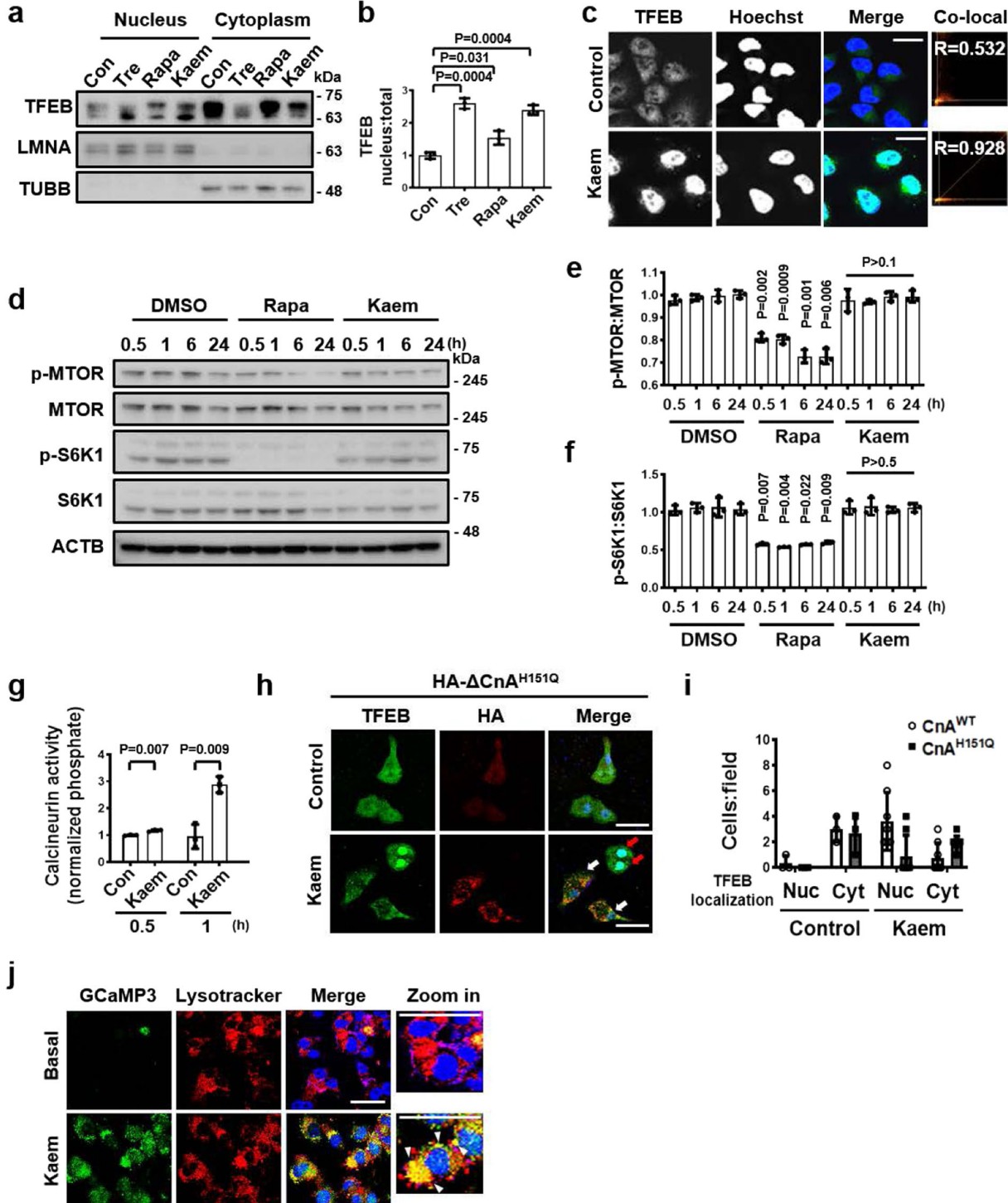

**Kaem promotes translocation of TFEB without MTOR inhibition.** To investigate how Kaem induces autophagy, we first examined whether Kaem upregulates the expression of autophagy-related genes. It is well-established that several transcription factors promote the expression of genes encoding autophagy- and lysosome-related proteins[34]. Remarkably, Kaem-induced translocation of TFEB into the nucleus, where the activated transcription factor promotes the expression of autophagy- and lysosome-related genes (Fig. 2a–c). TFEB is normally maintained in an inactive state via phosphorylation, which regulates the nuclear localization and export signals on the

transcription factor[35,36]. Among several cytosolic kinases and phosphatases known to regulate phosphorylation of TFEB, MTOR was reported as a major regulator of TFEB localization[36]. Therefore, inhibiting MTOR kinase activity is a well-established strategy for enhancing the transactivation of TFEB. To investigate the upstream molecular cascade associated with Kaem-induced TFEB activation, the total TFEB level in whole-cell lysates and the phosphorylation-level of MTOR and its substrate were investigated. The results of immunoblot analysis of TFEB in the lysate of Kaem-treated cells indicated a shift to the non-phosphorylated state[37] (Supplementary Fig. 4a). However, MTOR and its

**Fig. 2 Kaem induces TFEB translocation to the nucleus via Ca$^{2+}$ signaling regulation without MTOR inhibition. a, b** HeLa cells were treated with DMSO control, trehalose (Tre), rapamycin (Rapa), and Kaem for 6 h respectively, fractionated, and immunoblotted. Representative images (**a**) and intensity of nuclear TFEB immunoblot bands normalized to total TFEB (**b**). Kaem, 20 μM; Tre, 100 mM; Rapa, 10 μM. Graph shows mean ± SD from three independent experiments. **c** HeLa cells were treated with DMSO control or Kaem for 3 h. Confocal microscopy was conducted after immunostaining with the anti-TFEB antibody. Arrows indicate cells with nuclear translocation of TFEB. Co-localization (Co-local) analysis of nuclear TFEB using ImageJ2 (right). Kaem, 20 μM, Scale bar, 20 μm. **d–f** HeLa cells were treated with DMSO control, rapamycin (Rapa), and Kaem, respectively, for indicated period. Cell extracts were subjected to western blot analysis using antibodies against p-MTOR, MTOR, p-S6K1, and S6K1. Representative images (**d**), intensity of p-MTOR immunoblot bands normalized to MTOR **e**, intensity of p-S6K1 immunoblot bands normalized to S6K1 (**f**). Kaem, 20 μM; Rapa, 10 μM. Graph shows mean ± SD from three independent experiments. **g** HeLa cells were treated with DMSO control or Kaem for 0.5 or 1 h. Calcineurin activity assay was conducted according to the manufacturer's instructions. Absorbance ratio was measured at 620 nm. Kaem, 20 μM. Graph shows mean ± SD from three independent experiments. **h, i** HeLa cells were transfected with HA-ΔCnA-H151Q construct and treated with DMSO control or Kaem for 6 h. Confocal microscopy was performed after immunostaining with antibodies against HA (white arrows, HA-ΔCnA-H151Q-transfected cells showing no TFEB translocation by Kaem; red arrows, HA-ΔCnA-H151Q-untransfected cells showing TFEB translocation by Kaem) and TFEB (for endogenous TFEB). Representative images (**h**) and number of cells with nucleus (Nuc) or cytoplasmic (Cyt) TFEB localization per image field (**i**). Kaem, 20 μM, Scale bar, 50 μm. Graph shows mean ± SD ($n = 8$). **j** HeLa cells were transfected with GCaMP3-ML1 encoding a lysosome-specific Ca$^{2+}$ probe and then treated with DMSO control (Basal) or Kaem. Cells were stained with lysotracker and lysosomal Ca$^{2+}$ release was visualized by confocal microscopy. Kaem, 20 μM. Scale bar, 50 μm. Statistical significance was assessed by Student's $t$-test. ***$P < 0.001$; **$P < 0.01$; *$P < 0.05$.

substrate S6K1 remained in the phosphorylated state from the early (0.5 h) to late (24 h) phases of the experiment following Kaem treatment, whereas the known MTOR inhibitor rapamycin inhibited phosphorylation over this time course (Fig. 2d–f). These results suggest that alternative mechanisms involving factors other than MTOR regulate the TFEB signaling axis for autophagy induction of Keam.

One alternative means of activating TFEB without perturbing MTOR involves promoting de-phosphorylation by cytosolic phosphatases such as calcineurin, which is activated by cytoplasmic Ca$^{2+}$ ions[10]. Notably, Kaem enhanced calcineurin activity at 0.5 and 1 h, prior to TFEB translocation (Fig. 2g). To confirm the role of the phosphatase calcineurin in TFEB translocation induced by Kaem, HeLa cells were transfected with a dominant-negative mutant of the catalytic calcineurin A subunit (HA-ΔCnA-H151Q)[38]. Remarkably, Kaem-induced TFEB translocation into the nucleus was abolished in inactive calcineurin A-transfected cells (Fig. 2h–i and Supplementary Fig. 4b), indicating that calcineurin activation is pivotal in Kaem-induced TFEB translocation.

The lysosomal Ca$^{2+}$-calmodulin pathway is also required for maintaining MTOR activity[39]. As Kaem sustained MTOR activity (as shown in Fig. 2d–f), the lysosomal Ca$^{2+}$ channel was investigated to determine the source of transiently increased Ca$^{2+}$ levels in the cytoplasm. HeLa cells were transfected with GCaMP3-ML1 encoding a lysosome-specific Ca$^{2+}$ probe; the structure of the GFP tag on the channel protein in the outer lysosomal membrane shifts upon the capture of released Ca$^{2+}$ ions[40]. Kaem markedly induced lysosomal Ca$^{2+}$ release, as indicated by co-localization of green fluorescence with lysosome-associated red fluorescence (Fig. 2j), by contrast, pretreatment of lysosomotrophic compound glycyl-L-phenylalanine–β-naphthylamide (GPN) abolished Kaem-induced responses of GCaMP3-ML1 fluorescence[41] (Supplementary Fig. 4c). Collectively, these results demonstrate that Kaem-induced autophagy involves Ca$^{2+}$-/calcineurin-mediated TFEB signaling regulation.

**Kaem reduced the lipid content in adipocytes by enhancing autophagy.** Adipocytes have crucial roles in organisms for storing and maintaining energy sources. When high levels of glucose evoke insulin signaling, pre-adipocytes are fully differentiated into adipocytes to store energy in the form of LDs in the cells[42]. When cells become overloaded with LDs, there is an increased risk of insulin resistance, which is a major cause of metabolic syndromes such as obesity and diabetes[5,43]. Therefore, lipid degradation via autophagy is recognized as a crucial process in

lipid metabolism for maintaining organismal fitness. To investigate the role of Kaem in responding to metabolic stress in adipocytes through autophagy, we next explored the lipophagy activity of Kaem using 3T3-L1 cells. Kaem-treated 3T3-L1 cells were stained with AO to monitor lysosomal activity. The number of red puncta indicating acidic lysosomes increased in Kaem-treated cells compared to the control group (Fig. 3a, b). Moreover, the level of lysosomal membrane protein LAMP2A, which is reported as a key factor in autophagy flux[44,45], increased in Kaem-treated cells (Fig. 3c, d), indicating that Kaem enhances lysosomal function in 3T3-L1 cells.

To verify intact autophagy turnover in adipocytes, mRFP-GFP-LC3-transfected 3T3-L1 cells were treated with Kaem, which resulted in an increase in mRFP/GFP puncta (autolysosomes) (Fig. 3e, f). The 3T3-L1 cells were induced to differentiate into adipocytes via induction of insulin signaling to confirm autophagy flux. Levels of the autophagy markers p62 and LC3B declined upon Kaem treatment for 24 h in differentiated adipocytes (Fig. 3g–i). To confirm that the Kaem-induced reduction in LC3 level was due to autophagy-turnover, protein level was monitored for 0.5–24 h according to LC3-II degradative kinetics (Supplementary Fig. 5a). The reduction in LC3 and p62 expression by Kaem was abolished in the presence of the lysosomal V-ATPase inhibitor bafilomycin A1 or CQ, indicating that Kaem-promoted autophagic degradation (Fig. 3g–i and Supplementary Fig. 5b). These results suggest that Kaem markedly enhances autophagy in both pre- and differentiated 3T3-L1 adipocytes, suggesting a promising role of the compound in treating disorders of lipid metabolism.

Notably, Kaem-induced autophagy in 3T3-L1 cells was not associated with any perturbation of AMPK-MTOR signaling, consistent with our previous results in HeLa cells (Supplementary Fig. 5c). To explore the correlation between HeLa and 3T3-L1 cells, the role of TFEB in 3T3-L1 cells was also investigated. As an upstream regulator of TFEB de-phosphorylation, calcineurin exhibited increased activity following Kaem treatment in 3T3-L1 cells (Supplementary Fig. 5d). The level of TFEB protein expression increased from 0.5 h after Kaem treatment[46] (Supplementary Fig. 5a). LC3 exhibited a fluctuating expression pattern, which increased and decreased between 0.5 and 24 h after Kaem treatment, consistent with the time of increased TFEB expression (Supplementary Fig. 5a). Moreover, the levels of protein expression of other autophagy-related factors, such as those involved in autophagosome formation, lysosomal biogenesis, and autophagy sequestration (sqstm/p62), appeared to increase upon Kaem treatment,

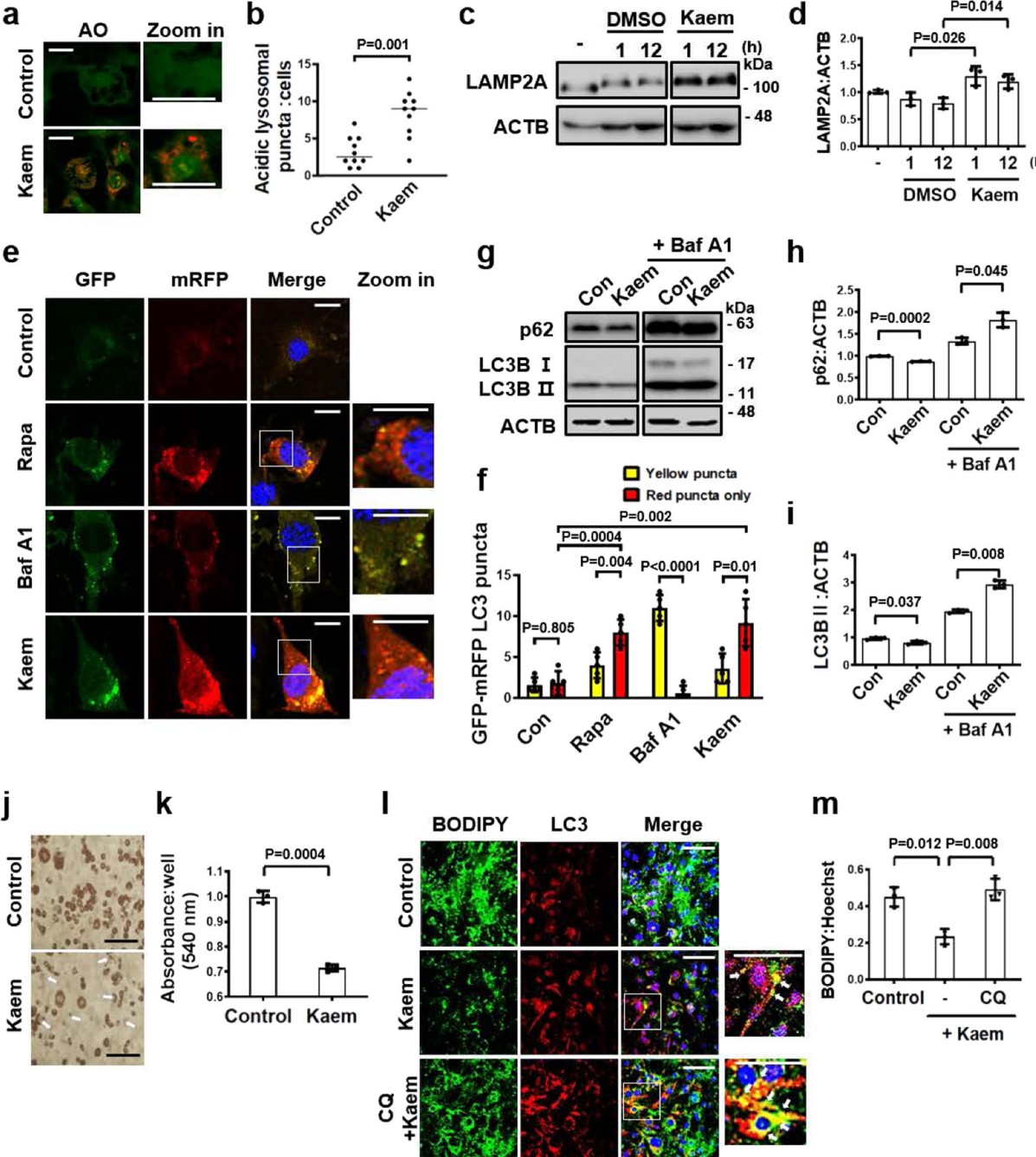

**Fig. 3 Kaem reduces lipid droplets by inducing autophagy flux in 3T3-L1 cells. a, b** 3T3-L1 cells were differentiated to adipocytes, treated with DMSO control or Kaem for 24 h. Cells were stained with acridine orange (AO) and examined by confocal fluorescence microscopy. Representative images (**a**) and acidic lysosome red puncta (**b**). Kaem, 20 μM. Scale bar, 50 μm. Graph shows mean ± SD ($n = 10$). **c, d** 3T3-L1 cells were treated with DMSO control or Kaem for the indicated period. Cell extracts were subjected to western blotting using an anti-LAMP2A antibody. Representative images (**c**) and LAMP2A immunoblot band intensity normalized to ACTB (**d**). The blots were processed in parallel. Kaem, 20 μM. Graph shows mean ± SD from three independent experiments. **e, f** 3T3-L1 cells transfected with mRFP-GFP-LC3 were treated with rapamycin (Rapa), bafilomycin A1 (Baf A1), or Kaem for 24 h, followed by confocal microscopy. Representative images (**e**) and number of yellow and red puncta (**f**). Kaem, 20 μM; Rapa, 10 μM; BafA1, 10 nM. Scale bar, 10 μm. Graph shows mean ± SD ($n = 5$). **g–i** Differentiated 3T3-L1 cells were treated with Kaem in the absence/presence of bafilomycin A1 (Baf A1). Cell extracts were subjected to western blot analysis using antibodies against LC3B and p62. Representative images (**g**) and p62 (**h**) and LC3B (**i**) immunoblot band intensity normalized to ACTB. The blots were processed in parallel. Kaem, 20 μM; Baf A1, 10 nM. Graph shows mean ± SD from three independent experiments. **j, k** 3T3-L1 cells differentiated for 9 days treated with Kaem three times from days 4 to 8. Cells were stained with oil-red-O (ORO) and examined by microscopy. Representative images (**j**) and ORO dye were extracted, and optical density was measured using a plate reader (**k**). Scale bar, 100 μm. Graph shows mean ± SD ($n = 3$). **l, m** 3T3-L1 cells differentiated for 5 days were treated with Kaem in the absence or presence of chloroquine (CQ). Confocal microscopy was performed after immunostaining with anti-LC3 antibody and BODIPY 493/503 staining. Representative images, white arrows indicate co-localization of lipid droplets with LC3 (**l**). BODIPY intensity was measured using ImageJ2 (**m**). Kaem, 20 μM; CQ, 10 μM. Scale bar, 50 μm. Graph shows mean ± SD. Statistical significance was assessed by Student's $t$-test. ***$P < 0.001$; **$P < 0.01$; *$P < 0.05$.

indicating that Kaem also activates autophagy at the transcription level in 3T3-L1 cells (Supplementary Fig. 5e).

To investigate whether Kaem-induced autophagy has a role in lipid metabolism, the effect on LD degradation was explored in adipocytes. Oil-Red-O staining of 3T3-L1 adipocytes revealed that LD accumulation decreased following Kaem treatment (Fig. 3j, k). To examine the relevance of this result with the induction of autophagic lipolysis, we investigated whether LDs decrease following inhibition of lysosomal activity using CQ. Following lysosomal inhibition by CQ, the decrease in lipid content induced by Kaem was reversed, with lipid accumulation. Moreover, immunostaining indicated that Kaem treatment increased co-localization LC3 puncta and LDs (Fig. 3l, m and Supplementary Fig. 5f, g). These results suggest that Kaem promotes LD degradation in an autophagic degradation-dependent manner known as lipophagy[47,48]. In addition, a previous report demonstrated that TFEB transcriptionally drives browning of inguinal white adipose tissue by enhancing the expression of uncoupling protein 1 (UCP1), a key thermogenic protein[49]. In 3T3-L1 adipocytes, the UCP1 protein level increased slightly following Kaem treatment, comparable to treatment with the β-adrenergic agonist norepinephrine, suggesting a role for Kaem as a chemical thermogenic stimulator (Supplementary Fig. 5h).

**Kaem-induced autophagy ameliorates metabolic dysregulation in vivo**. Next, we examined whether autophagy induction by Kaem improves the metabolic profile in diet-induced obese mice in vivo. Mice fed a high-fat diet were treated with vehicle or 10 mg/kg of Kaem via intraperitoneal injection every 2 days for 2 months. The bodyweight of Kaem-treated mice increased less than vehicle-treated mice (Fig. 4a–c), and the non-fasting blood glucose level was improved on the final day of the experiment (day 61) (Fig. 4d).

Fatty liver conversion, which is commonly associated with metabolic syndrome[50], was ameliorated by administration of Kaem in diet-induced obese mice, accompanied by autophagy induction in the liver tissue (Fig. 4e–h). The results of immunostaining indicated a reduction in the average size of LDs in the liver, with increased LC3 puncta, although the number of LDs did not decrease and may have even increased slightly, perhaps due to fractionation of the LDs. Additionally, LC3 surrounding the surface of the LDs increased (Fig. 4i–l, white arrows in Fig. 4i). These results indicate that Kaem induces autophagy, leading to LD degradation in fatty liver.

In addition, the total volume of subcutaneous fat tissue and visceral fat tissue were markedly reduced in Kaem-treated mice (Supplementary Fig. 6a, b). Kaem-treated mice exhibited autophagy activation in both intrascapular brown adipose tissue (iBAT), with increased LC3 levels, and visceral white adipose tissues (WAT), with decreased p62 levels (Supplementary Fig. 6c–h). In iBAT in particular, the expression of UCP1, a marker of active thermogenesis as well as TFEB target gene[51], was also increased by Kaem treatment, consistent with previous results indicating that Kaem induces UCP1 expression in adipocytes (Supplementary Fig. 5h). Immunoblotting of WAT also revealed a slight reduction in the expression of surface perilipin 1 on LDs in WAT of Kaem-treated mice, indicating LD degradation, although the observed difference was not meaningful due to variations between the animals in the group[52].

**Identification of mitochondrial TUFM as a direct binding target protein of Kaem**. To explore the underlying mechanism of the Kaem-mediated induction of autophagy, target proteins that directly interact with Kaem were identified by employing a recently developed method combining DARTS and LC–MS/MS analysis[53]. After the compound binds to target proteins with high

affinity, their structures undergo a shift that produces a conformational change. The stability of the target proteins can change due to the conformational change, leading to a change in susceptibility to proteolysis by the enzyme pronase. The levels of peptides of proteins exhibiting a change in proteolytic stability are determined using a quantitative LC–MS/MS analysis to identify target protein candidates. By a quantitative analysis of the total peptide pool, 10 target candidates exhibiting increased stability were identified (Fig. 5a and Supplementary Fig. 7a, b). By evaluating different autophagic pathways (conventional, endosomal, or alternative)[54,55], we found that Kaem induced the ATG7-mediated conventional autophagy pathway with p62 degradation. Kaem did not induce the RAB9-mediated alternative or RAB5-mediated endosomal autophagy pathways (Supplementary Fig. 8a, b). Based on these analyses, TUFM was finally selected as a target candidate since an autophagy-related role for TUFM was reported in studies involving recruitment of the ATG12-5-16L1 complex, which requires ATG7 to form the structural complex[56,57].

DARTS immunoblotting confirmed direct physical binding between Kaem and TUFM, which was absent with voltage-dependent anion-selective channel 1 (VDAC1), a mitochondrial negative control protein. The direct interaction of Kaem with TUFM enhanced the stability of TUFM against pronase treatment (Fig. 5b–c). Moreover, Kaem stabilized TUFM in a dose-dependent manner, indicative of a high-affinity molecular interaction between the compound and target protein (Fig. 5d–f).

Notably, the interaction between Kaem and TUFM saturated above $100\,\mu M$ of Kaem under pronase treatment at $1\,\mu g/mL$, indicating a binding affinity, based on an $EC_{50}$ concentration of Kaem, of $14.4\,\mu M$. These results strongly demonstrated that Kaem directly interacts with TUFM in vitro at a concentration comparable to that in living cells in vivo.

We next explored chemical structures in the physical interaction between Kaem and TUFM using proteome databases. Analysis of the peptide sequences of TUFM recovered by Kaem in the DARTS LC–MS/MS analysis revealed that six peptides (ⓐ–ⓕ) were specifically stabilized more than 10% compared to pronase alone treatment. These peptides were located on the translational GTPase (tr-type G) domain of the protein, indicating that this domain could be a binding site of Kaem (Fig. 5g). In consistent with these results, in silico docking analyses indicated that Kaem directly binds to the GTPase domain via interaction with G172, K256, and R421. Notably, G172 and K256 are located on stabilized peptides ⓒ and ⓔ, respectively (Fig. 5h, blue circles), although R421 is not located on one of the stabilized peptides (Fig. 5h–i). Thus, these results suggest that G172 and K256 residues might contribute to the key binding motif for Kaem and TUFM by enhancing the resistance of the protein to pronase digestion.

**TUFM interacts with the ATG12–ATG5 complex in Kaem-induced autophagy**. To investigate the biological relevance of TUFM in Kaem-induced autophagy, we examined whether Kaem induces TUFM to promote autophagosome formation by recruiting ATG family proteins, as the target protein was selected based on Kaem's dependence on the ATG7 regulatory pathway among the several autophagy types. Following Kaem treatment, immunocytochemistry analysis indicated that TUFM co-localized with ATG12, and co-immunoprecipitation analysis indicated direct interaction with the ATG12–ATG5 complex (Fig. 6a, b). Erlotinib, a well-known inhibitor of EGFR, also promoted this protein interaction, as reported previously[58] (Fig. 6a, b). These results indicate that Kaem promotes interaction between TUFM and the ATG12–ATG5 complex to enhance autophagy, particularly promoting autophagosome formation.

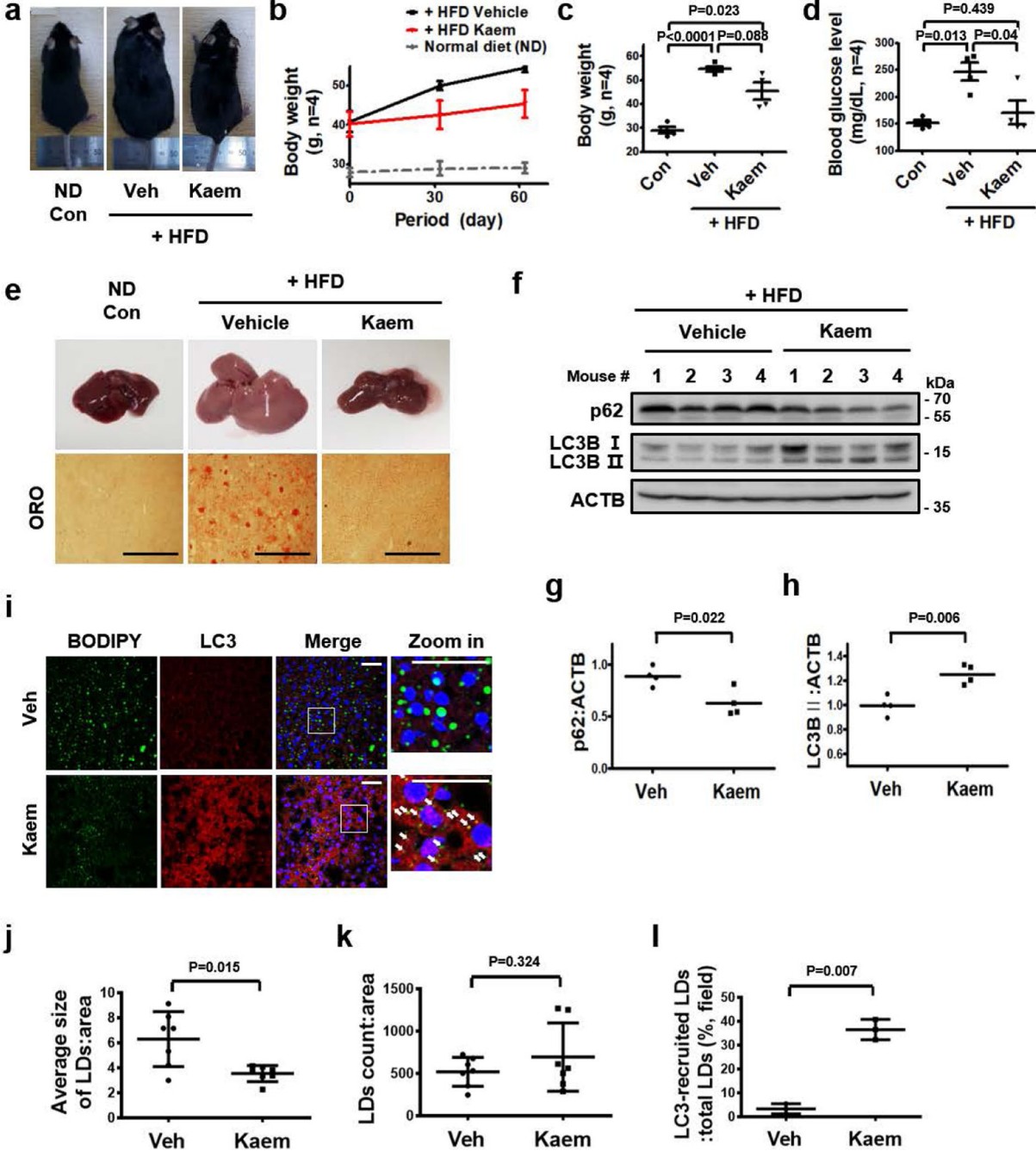

**Fig. 4 Kaem ameliorates obesity in mice fed a high-fat diet (HFD).** C57BL/6j mice were fed an HFD for 2 months and treated with vehicle (Veh) or Kaem every 2 days. **a** Representative photographs of mice ($n = 4$). Intraperitoneal injection, Kaem, 10 mg/kg. **b** Bodyweight change during Kaem treatment. Graph shows mean ± SEM ($n = 4$). **c** Bodyweight on the final day (day 61). Graph shows mean ± SD ($n = 4$). **d** Non-fasting blood glucose level on the final day (day 61). Graph shows mean ± SD ($n = 4$). **e** After each mouse was sacrificed, the liver was removed. Representative photographs of the liver (upper). Tissues were sectioned at the 10-μm thickness, and microscopy was performed after staining with oil-red-o (ORO) (below). Scale bar, 1 mm. **f–h** Each mouse liver tissue lysate was subjected to western blot analysis using antibodies against p62 and LC3B. Representative images (**f**) and p62 (**g**) and LC3B (**h**) immunoblot band intensity normalized to β-actin. Graphs show mean ± SD ($n = 4$). **i–l** Liver tissues were sectioned at 10-μm thickness, and confocal microscopy was performed after immunostaining with anti-LC3 antibody and BODIPY 493/503 staining. Representative images, arrows indicate encircling of lipid droplets with LC3 (**i**). Average size of LDs (**j**), number of LDs (**k**), and (%) of LC3-recruiting LDs (**l**). Scale bar, 50 μm. Graphs show mean ± SD. Statistical significance was assessed by Student's *t*-test. ***$P < 0.001$; **$P < 0.01$; *$P < 0.05$.

**TUFM is required for Kaem-induced autophagic turnover.** To verify TUFM as a target protein of Kaem, the biological effects of Kaem were investigated in the case of TUFM deficiency via genetic knockdown. Remarkably, the functional reduction of TUFM using silencing RNA abolished Kaem-induced autophagy

(Fig. 7a, b), suggesting that Kaem requires TUFM as a binding partner to induce intact autophagy flux. Because Kaem-promoted LD degradation via autophagy (Fig. 3l, m), we further examined the TUFM dependence of Kaem-mediated degradation of LDs. In HepG2 cells loaded with a combination of palmitic acid and oleic

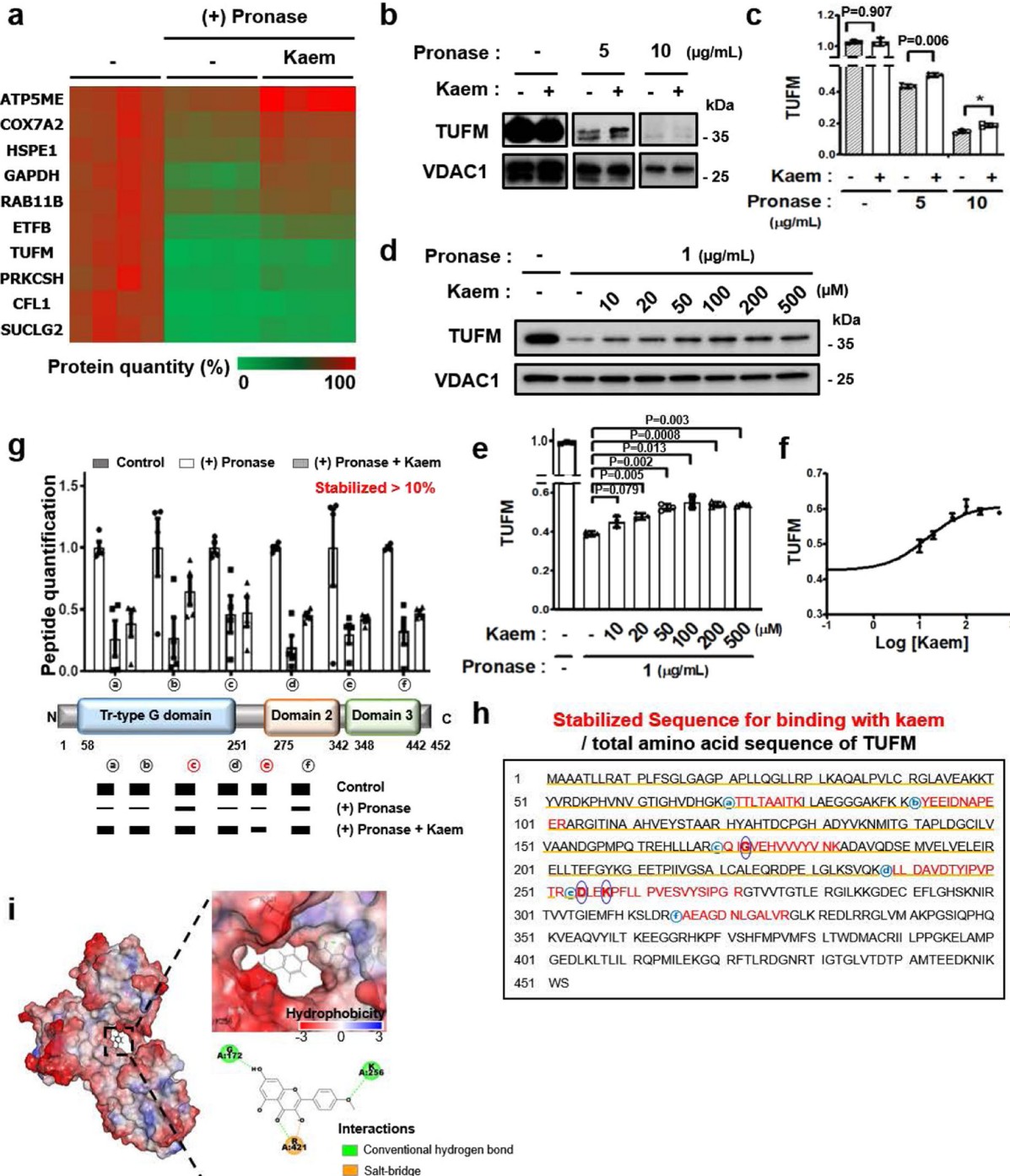

**Fig. 5 DARTS LC–MS quantitative proteomic analysis indicates TUFM is a biophysical target of Kaem. a** Heatmap of top 10 proteins showing significantly increased stability to proteolysis upon Kaem treatment. 3T3-L1 cell lysate was treated with pronase for 10 min with/without Kaem pretreatment. SWATH analysis was conducted to identify proteins with varying protease stability (by KBSI). Top 10 protein target candidates were selected according to the sequential criteria based on the amount of detected peptide. **b**, **c** DARTS assay for target validation. 3T3-L1 cell lysate was treated with pronase for 10 min with/without Kaem pretreatment and subjected to western blot analysis using anti-TUFM and anti-VDAC1 antibodies. Representative images (**b**) and intensity of TUFM immunoblot bands (**c**). The blots were processed in parallel. Kaem, 2 mM. Graph shows mean ± SD from three independent experiments. **d**–**f** 3T3-L1 cells were treated with pronase for 10 min with/without Kaem pretreatment in a dose-dependent manner, then subjected to western blot analysis using anti-TUFM and anti-VDAC1 antibodies. Representative images (**d**), the intensity of TUFM immunoblot bands (**e**), and sigmoidal curve (**f**). Graph shows mean ± SD from three independent experiments. **g** Graph of top isotope quantitation (TIQ) for TUFM peptides (upper). Schematic illustration of the peptide-locus of TUFM consisting of the tr-type G domain, domain 2, and domain 3 (lower). Graph shows mean ± SEM of each peptide quantification (n = 4). **h** Complete amino acid sequence of TUFM. Core amino acids correlated with in silico docking analysis are indicated by blue circles. **i** In silico docking model of Kaem directly interacting with TUFM (PDB ID: 1XB2, interaction energy: 44.9 kcal/mol). Presumably binding amino acids and interaction mode are depicted. Statistical significance was assessed by the Student's t-test. ***P < 0.001; **P < 0.01; *P < 0.05.

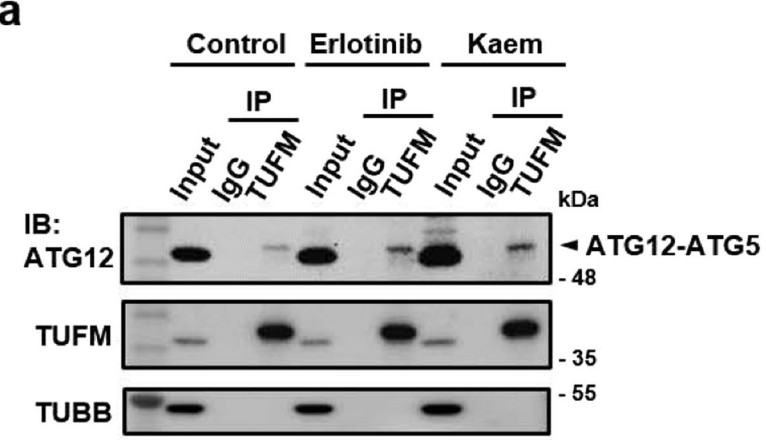

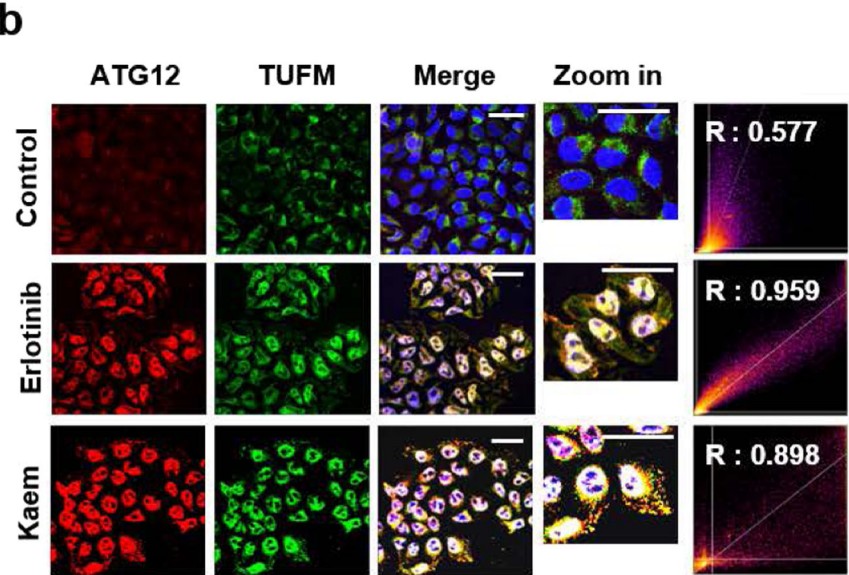

**Fig. 6 Direct interaction between Kaem and TUFM promotes a protein-protein interaction related to autophagic phagophore formation. a** HepG2 cells were treated with erlotinib or Kaem for 24 h. Co-immunoprecipitation (Co-IP) was conducted using anti-TUFM antibody followed by western blotting. Erlotinib, 20 µM, Kaem, 20 µM. **b** HeLa cells were treated with erlotinib or Kaem for 24 h. Confocal microscopy was conducted after co-immunostaining with anti-ATG12 and anti-TUFM antibodies. Co-localization analysis of ATG12 and TUFM using ImageJ2 (right). Scale bar, 50 µm. Erlotinib, 20 µM, Kaem, 20 µM.

acid to stimulate LD synthesis and then treated with Kaem, the accumulation of BODIPY-stained LDs decreased, indicating LD clearance mediated by Kaem. However, the LD clearance effect of Kaem was abolished following genetic knockdown of TUFM in cells transfected with silencing RNA (Fig. 7c, d).

**Increasing TUFM levels enhances autophagy and promotes lipid degradation**. As our results indicated that TUFM has an independent role in metabolic regulation, the effects of TUFM on autophagy activity and lipid degradation were evaluated under conditions of genetic overexpression. Increased levels of TUFM enhanced autophagy, including p62 degradation and LC3-II conversion, similar to Kaem treatment (Fig. 7e–g and Supplementary Fig. 9a, b). Overexpression of TUFM alone enhanced LD clearance in Huh7 cells, although there was a reduction in the number of LDs in each cell, the average size of the LDs in each cell declined non-meaningfully (Fig. 7h–j). These results

demonstrate that TUFM alone has a role in metabolic regulation by enhancing autophagy.

**Kaem modulates mtROS and ETC components to facilitate autophagy with TUFM**. TUFM was identified and validated as a biologically relevant target in Kaem-induced autophagy. However, as Kaem promotes TFEB translocation to the nucleus to induce autophagy, the link between the target protein TUFM and TFEB translocation remains to be elusive. Because we found that Kaem induces lysosomal $Ca^{2+}$ ion release, we hypothesized that TUFM indirectly regulates lysosomal ion channels. Previous studies demonstrated that mtROS regulate the lysosomal $Ca^{2+}$ channel TRPML1, which binds to calmodulin, leading to calcineurin activation[40], suggesting a change in the channel's redox state[59,60]. Thus, we examined whether Kaem regulates the redox state of TRPML1 to promote $Ca^{2+}$ ion release. As mtROS levels increased slightly with Kaem treatment without severe mitochondrial toxicity (Supplementary Fig. 10a–f), we examined the

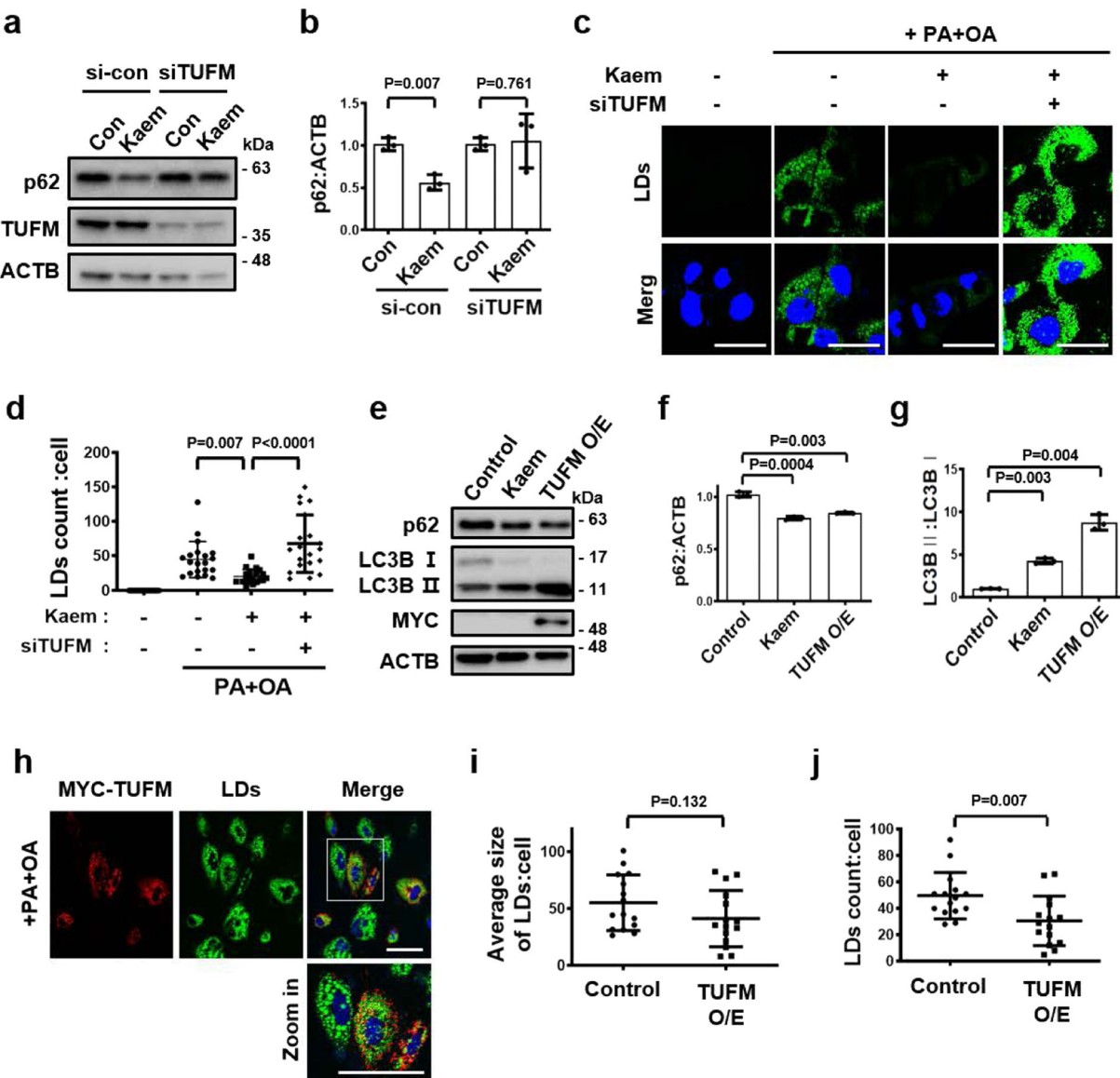

**Fig. 7 Kaem promotes autophagy-mediated LD degradation by directly interacting with mitochondrial EF-Tu. a**, **b** Huh7 cells were transfected with/without siRNA targeting TUFM for 24 h, followed by Kaem treatment for 24 h. Cell extract was subjected to western blot analysis using antibodies against p62 and TUFM. Representative images (**a**) and intensity of p62 immunoblot bands normalized to ACTB (**b**). Kaem, 20 μM. Graph shows mean ± SD from three independent experiments. **c**, **d** HepG2 cells were transfected with/without siRNA targeting TUFM for 24 h then treated with palmitic acid and oleic acid (PA + OA) for lipid droplets (LDs) loading for 24 h, followed by treatment with Kaem for 24 h. Confocal microscopy was performed after BODIPY 493/503 staining. Representative images **c** and number of lipid droplets in each cell (**d**). Kaem, 20 μM, PA, 400 μM, OA, 800 μM. Scale bar, 20 μm. Graph shows mean ± SD (n = 20). **e–g** Huh7 cells were treated with Kaem or transfected with TUFM construct (MYC/DDK tagged, for overexpression (O/E)) for 24 h. Cell extract was subjected to western blot analysis using antibodies against p62, LC3B, and MYC. Representative images (**e**) and intensity of p62 (**f**) and LC3B (**g**) immunoblot bands normalized to ACTB. Kaem, 20 μM. Graph shows mean ± SD from three independent experiments. **h–j** Huh7 cells were transfected with TUFM (MYC/DDK tagged) construct for overexpression (O/E), followed by PA + OA treatment for 24 h. Confocal microscopy was performed after BODIPY 493/503 staining. Representative images (**h**) and size (**i**) and number of lipid droplets (**j**) in each cell. Kaem, 20 μM, PA, 400 μM, OA, 800 μM. Scale bar, 50 μm. Graph shows mean ± SD (n = 15). Statistical significance was assessed by the Student's t-test. \*\*\*P < 0.001; \*\*P < 0.01; \*P < 0.05.

effect of mtROS on lysosomal $Ca^{2+}$ release following Kaem treatment. Kaem treatment alone promoted lysosomal $Ca^{2+}$ release and exhibited sequential lysosomal exocytosis requiring lysosomal $Ca^{2+}$ ion flux. However, following mtROS scavenging with mitoTempo, Kaem-induced lysosomal $Ca^{2+}$ efflux, and exocytosis were markedly reduced, suggesting that mtROS are required for Kaem-induced autophagy (Supplementary Figs. 10a, b and 11a). In addition, Kaem-promoted TFEB translocation to the nucleus was suppressed by co-treatment with chemical

inhibitors that act on the mtROS-TRPML1-calcineurin cascade: ROS (cellular ROS scavenger *N*-acetyl-ʟ-cysteine, NAC), lysosomal $Ca^{2+}$ release (TRPML1 synthetic inhibitor 1, ML-SI1), and calcineurin inhibitors tacrolimus (FK506) with cyclosporin A (CsA), but translocation was not suppressed by co-treatment with the NOX inhibitor DPI (Fig. 8a, b).

To examine whether these results were in line with the Kaem-induced autophagy regarding each factor in the cascade, other chemical inhibitors were utilized to determine the relevance of

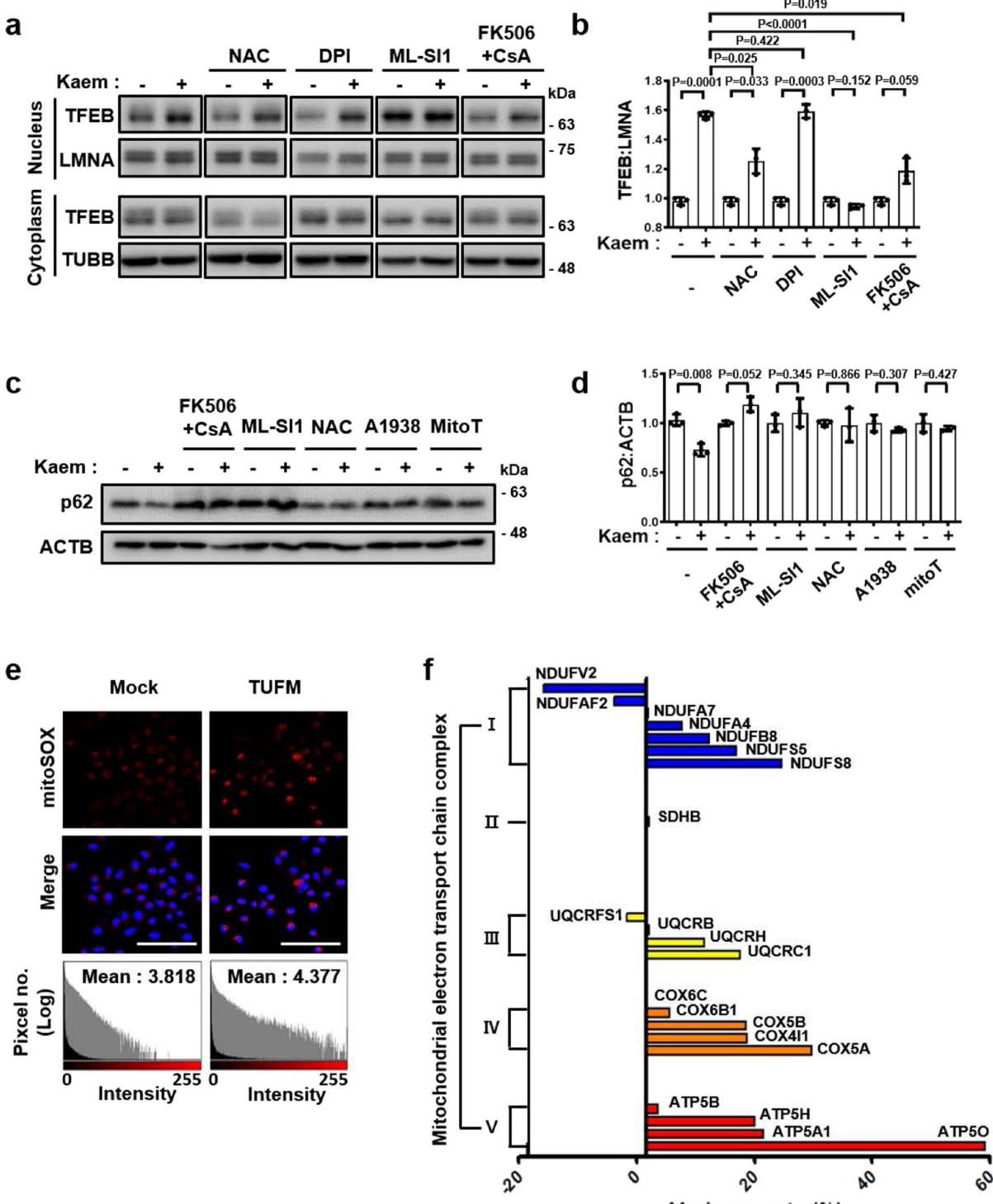

**Fig. 8 Kaem enhances autophagy via mtROS. a, b** HeLa cells were treated with Kaem with/without NAC or DPI or ML-SI1 or FK506 + CsA for 3 h, then fractionated and immunoblotted. Representative images **a** and intensity of nuclear TFEB immunoblot bands normalized to LMNA (**b**). Kaem, 20 μM; NAC, 5 mM; DPI, 1 μM; ML-SI1, 25 μM; FK506, 5 μM; CsA, 10 μM. Graph shows mean ± SD from three independent experiments. The samples derive from the parallel experiments and the blots were processed in parallel. **c, d** 3T3-L1 cells were treated with Kaem with/without FK506 + cyclosporine A (CsA) or ML-SI1 or NAC or A1938 or mitoTempo (MitoT). Cell extract was subjected to western blot analysis using antibodies against p62. Representative images (**c**) and intensity of p62 immunoblot bands normalized to ACTB (**d**). Kaem, 20 μM; FK506, 5 μM; CsA, 10 μM; ML-SI1, 25 μM; NAC, 5 mM; A1938, 10 μM; mitoTempo, 100 μM. Graph shows mean ± SD from three independent experiments. **e** Huh7 cells were transfected with TUFM for 24 h, stained with mitoSOX, and examined by confocal microscopy (upper). Fluorescence intensity histogram generated using ImageJ 2 (lower). Scale bar, 100 μm. **f** 3T3-L1 cells were treated with Kaem for 24 h. Mitochondrial proteins were fractionated and subjected to LC–MS/MS analysis after TMT labeling. The graph shows mitochondrial ETC protein levels ($n = 1$). Statistical significance was assessed by Student's $t$-test. ***$P < 0.001$; **$P < 0.01$; *$P < 0.05$.

mtROS: mitochondrial electron transport chain (ETC) complex III component UQCRB inhibitor A1938[61], and the mtROS-specific scavenger mitoTempo. Degradation of autophagy substrate p62 was noted with Kaem treatment alone, which was abolished by co-treatment with inhibitors of mitochondrial or cellular ROS, lysosomal $Ca^{2+}$ release, or TFEB translocation, but not by inhibition of the cytoplasmic ROS-generating protein NOX (DPI) (Fig. 8c, d and Supplementary Fig. 11b, c). These results demonstrate that Kaem-induced autophagy is dependent on the respective factors in the mtROS-TRPML1-TFEB cascade.

To elucidate the direct link between TUFM and mtROS generation, TUFM was over-expressed in Huh7 cells, which normally express low levels of TUFM. Notably, TUFM over-expression ($500 \, ng/5 \times 10^4$ cells) slightly enhanced mtROS generation (1.18-fold increase in mtROS: Hoechst intensity compared with control) (Fig. 8e), whereas mtROS generation was reduced in cells transfected with a higher amount of TUFM ($1000 \, ng/5 \times 10^4$ cells), suggesting that TUFM has a role in mtROS regulation. To elucidate the mechanism underlying the regulatory effect of the TUFM modulator Kaem on mtROS generation, the mitochondrial proteome was analyzed to assess fluctuations in the expression levels of ETC components upon Kaem treatment. Quantitative proteomics analysis revealed variations in mitochondrial protein expression levels following Kaem treatment (Fig. 8f and Supplementary Fig. 12). These results demonstrate that Kaem modulates the expression of various mitochondrial proteins, which may be the primary mechanism underlying the effect of Kaem on mtROS generation.

## Discussion

This study identified the natural compound kaempferide (Kaem) as a discovered autophagy enhancer that improves the metabolic condition in diet-induced obese mice. Since Kaem was identified via phenotypic screening, its biological activity was investigated to elucidate how it functions in vivo in order to avoid potential adverse side effects in clinical use. DARTS, a method based on changes in protease susceptibility, was applied to identify the target proteins of Kaem in combination with LC–MS/MS analysis. Using this approach, we identified TUFM as a key direct target protein of Kaem based on biophysical interaction and biological relevance as well. The DARTS results indicated that the 50% resistance recovery concentration was $14.2 \, \mu M$, which is similar to the in vivo working concentration of $20 \, \mu M$ (Fig. 5d–f), indicating the reliability of the DARTS method. However, the possibility of other binding partners having a role in the biological effect must be considered in order to fully elucidate the biological mechanism of this potentially valuable chemical probe.

We hypothesized that TRPML1 is directly regulated by Kaem, as our data showed that lysosomal $Ca^{2+}$ was immediately released upon Kaem treatment. Hence, the binding affinity of TRPML1 with Kaem was confirmed using DARTS analysis, which indicated a weaker binding affinity ($>500 \, \mu M$) than of TUFM (Supplementary Fig. 13). These results suggest that Kaem directly regulates TUFM with much higher molecular affinity than TRPML1. In addition, LC–MS/MS analysis enables the identification of potential multiple target proteins for a given compound. Of the 10 candidates identified as shown in Fig. 5a, the nine proteins other than TUFM could also be binding partners of Kaem, albeit we focused on TUFM as a key biologically relevant target protein in this study based on biophysical and biological relevancy of the candidate proteins. Given the results of "regulation of mtROS" by Keam, we further investigated mitochondrial ETC proteins, cytochrome c oxidase subunit 7A2 (COX7A2) or ATP synthase subunit e (ATP5ME) among the target candidates. Reduction of ATP5ME expression level by assessing

silencing RNA enhanced autophagy with p62 degradation 0.91-fold, where the variation of the protein expression did not affect the enhancement of mtROS (Supplementary Fig. 14a–d). COX7A2, on the other hand, enhanced autophagy with p62 degradation 0.89-fold conveying upregulation of mtROS 1.16-fold under overexpression condition (Supplementary Fig. 14a, b, e, f), thus indicating that COX7A2 could be suggested as another target protein in respect with Kaem-induced autophagy. However, in the comparison between TUFM and COX7A2, autophagy activity of TUFM overexpression was much mimic chemical-action of Kaem, which exhibited p62 degradation 0.82-fold and 0.78-fold, respectively (Fig. 7e, f), rather than COX7A2 overexpression that showed a minor effect on autophagy. Although COX7A2 may partially contribute to affecting the biological activities of Kaem, these results demonstrate that TUFM is a highly relevant target protein for Kaem-induced autophagy and LD degradation activities.

Kaem belongs to the flavonol subgroup of flavonoids that features double bond and hydroxyl group in C ring. Kaempferol (Krol), which is famously described as a beneficial ingredient in many studies[62], also belongs to the subgroup (Supplementary Fig. 15a). Based on the structural similarity between these compounds Kaem and Krol, except methoxyl and hydroxyl group on the B ring of each compound, we assessed structure–target binding activity-relationship (SAR) analysis via DARTS assay. Compared to the prior result of TUFM stabilization by Kaem in a dose-dependent manner shown in Fig. 5d–f, Krol maintained TUFM stabilization with a weaker affinity (Saturation-$EC_{50}$ concentration of $74.0 \, \mu M$) (Supplementary Fig. 15b, c). This result indicated that the methoxyl group on B ring is required to bind to TUFM with high affinity. From the primary SAR information, other compounds in different flavonoid-subgroups were further investigated whether subgroup structures and methoxyl group in each group is correlated (Supplementary Fig. 15a). Remarkably, compounds in the isoflavone subgroup did not show any binding affinity to TUFM, and the methoxyl group is pivotal to bind to TUFM regardless of to which subgroups compounds belong (Supplementary Fig. 15d–f). This SAR result provides better information on how to regulate TUFM in a specific manner.

Our study revealed that TUFM has a role in improving metabolic dysregulation. These results are in line with other reports suggesting a genetic association between TUFM and energy metabolism. For example, Gonzalez et al.[63] reported that genomic inversion at 16p11.2, in which TUFM is located proximally, protects against the joint occurrence of asthma and obesity. They revealed that the TUFM expression level is upregulated under the inversion, indicating an association between higher TUFM expression and improved basal energy balance[63]. Another study explored obesity-associated single-nucleotide polymorphisms associated with CpG methylation and reported that the TUFM gene is regulated by promoter methylation in obesity[64], suggesting that exposure to a metabolic burden could have unfavorable consequences for TUFM with negative effects on expression that lead to a vicious cycle of metabolic dysregulation. In addition, other studies indicated the relevance of TUFM in maintaining mitochondrial function during exercise in obesity[65] and its association with the insulin cascade[66]. Although these studies suggested TUFM has an important metabolic role, they did not provide direct evidence of TUFM modulation. The present study demonstrated the role of TUFM in metabolic regulation via both chemical and genetic modulation.

An important question is whether the binding between Kaem and TUFM is related to the observed strong increase in $Ca^{2+}$ signaling activation. TUFM is known to have multiple roles in regulating mitochondrial ETC components[67], inducing

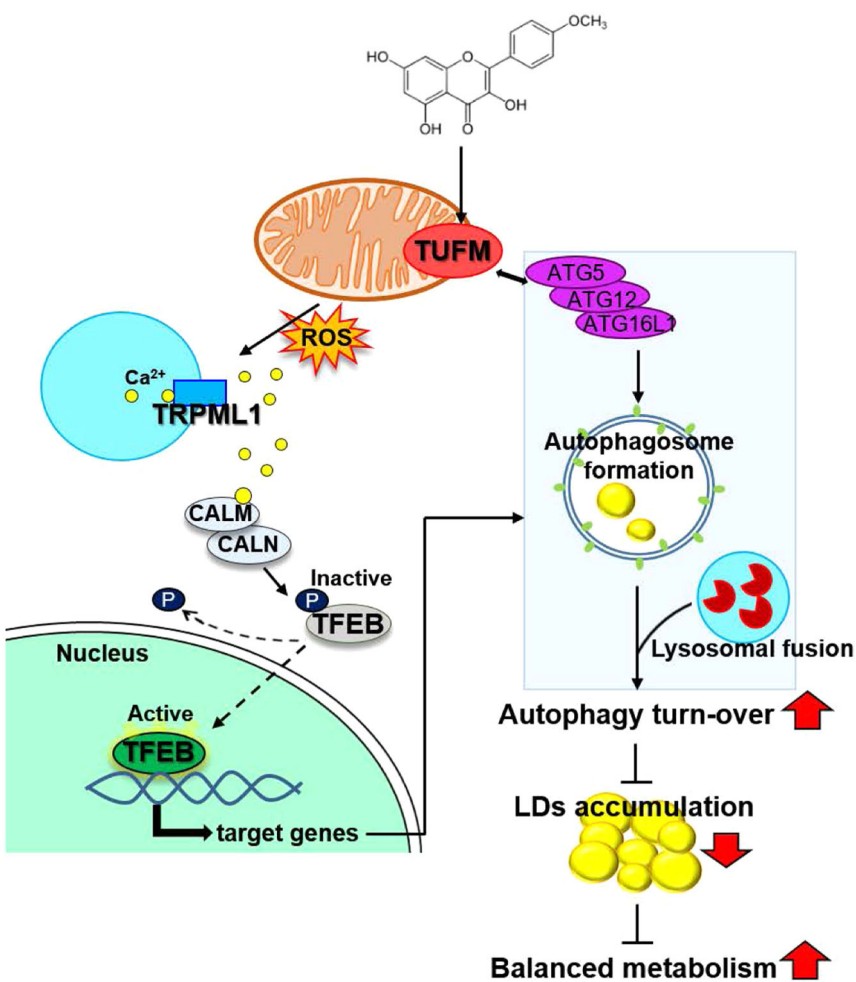

**Fig. 9 Schematic illustration summarizing the mechanism of Kaem-induced autophagy.** Kaem regulates TUFM to induce autophagy in a coordinated manner: (i) promoting autophagosome formation (right); and (ii) activating autophagy master regulator (left). Chemical structure of Kaem (top). ATGs, autophagy related genes; LDs, lipid droplets; ROS, reactive oxygen species; CALM, calmodulin; CALN, calcineurin; TFEB, transcription factor EB.

autophagy by recruiting the ATG12-5-16L1 complex, and inhibiting inflammation by binding to NLRX1[56,58]. A plausible model supported by our data is that mtROS generated by the mitochondrial ETC may activate TRPML1 by regulating the oxidative state of the channel[59,60]. We, therefore, examined the protein levels of mitochondrial ETC components by LC–MS/MS and found that several components were markedly upregulated following Kaem treatment (Fig. 8f, Supplementary Fig. 12). This could cause an imbalance in mitochondrial respiration and/or facilitate electron leakage, leading to the generation of ROS[68]. Although the direct role of TUFM in mitochondrial protein translation remains to be explored in the future study, our results demonstrated that Kaem transiently induces mtROS generation, which leads to enhanced TFEB translocation and autophagy (Fig. 9). Collectively, TUFM is identified as a biologically relevant target protein by which Kaem induces autophagy to improve the overall metabolic condition. In addition, these results also provide an insight into the potential role of TUFM in treating metabolic syndrome.

## Methods

**Materials**. A chemical library of 658-natural compounds was kindly provided by Dr. Sang Jeon Chung of Sungkyunkwan University (Suwon, Korea). Kaempferide (69545), dimethylsulfoxide (D2650), bafilomycin A1 (B1793), rapamycin (553210), tiliroside (79257), chloroquine (C6628), orlistat (O4139), palmitic acid (P5585), oleic acid (O1383), acridine orange (A6014), oil-red-O (O0625), dexamethasone (D8893), insulin (I0516), and 3-isobutyl-1-methylxanthine (I5879) were purchased

from Sigma-Aldrich. BODIPY 493/503 (D3922), Hoechst33342 (H3570), lipofectamine LTX (94756), lipofectamine 2000 (52887), Plus reagent (10964), protease and phosphatase inhibitor solution (78441), M-PER kit (89842Y), DMEM, fetal bovine serum (FBS), bovine serum, and antibiotics were purchased from Invitrogen ThermoFisher Scientific. For in vivo experiments, Kaempferide (K0057) was purchased from TCI Chemicals. siRNA targeting TUFM was purchased from Dharmacon. mRFP-GFP-LC3B plasmids were kindly provided by Dr. Jaewhan Song of Yonsei University (Seoul, Korea).

**Cell culture and treatment**. 3T3-L1 pre-adipocytes were cultured in DMEM supplemented with bovine serum (10% v/v) and penicillin/streptomycin (1% v/v) and used at passages 13–18. To induce differentiation of 3T3-L1 cells into fully differentiated adipocytes, the cells were incubated in MDI medium containing 3-isobutyl-1-methylxanthine, dexamethasone, and insulin for 2 days, after which they were incubated in insulin medium for 2 days and then DMEM supplemented with FBS (10% v/v) and penicillin/streptomycin (1% v/v). HeLa, HepG2, and Huh7 cells were cultured in DMEM supplemented with FBS (10% v/v) and penicillin/streptomycin (1% v/v). All cell cultures were maintained at pH 7.4 in a humidified incubator at 37 °C in a 5% (v/v) $CO_2$ atmosphere.

**mRFP-GFP-LC3B plasmid transfection**. Cells were transfected with the mRFP-GFP-LC3B plasmid using lipofectamine LTX transfection reagent (Invitrogen) for 24 h. The cells were treated with drug (DMSO control, rapamycin, baf A1, and Kaem) for 24 h. Nuclei were stained with Hoechst. Following incubation for 20 min, the cells were fixed with 4% formaldehyde and washed three times with PBS. Images were obtained using an LSM880 confocal microscope at ×400 magnification. Red and green puncta were then counted.

**Immunoblotting and co-immunoprecipitation (co-IP)**. Soluble proteins were harvested from cells using SDS lysis buffer (50 mM Tris-HCl [pH 6.8] containing 10% glycerol, 2% SDS, 10 mM dithiothreitol, and 0.005% bromophenol blue).

Equal volumes of protein were separated by 10% or 12.5% SDS-PAGE, and the proteins were then transferred onto polyvinylidene fluoride membranes (EMD Millipore, Billerica, MA, USA). Blots were then blocked and immunolabeled overnight at 4 °C with the following primary antibodies; anti-LC3B (Cell Signaling Technology), anti-LAMP2A, anti-VDAC1, anti-actin (Abcam), anti-p62 (BD Biosciences, Franklin Lakes, NJ, USA), anti-TUFM (Atlas Antibodies), and anti-TRPML1 (Novus). Immunolabeling was visualized using an enhanced chemiluminescence kit (Amersham Life Science, Inc., Amersham, UK) according to the manufacturer's instructions. Images were quantified using Image Lab (Bio-Rad, Hercules, CA, USA). ACTB was used as an internal control. All band intensity values are proportional to the amount of target protein on the membrane within the linear range of detection. For co-IP, cells treated with indicated compounds were harvested and lysed with IP-lysis buffer (50 mM Tris-HCl [pH 7.8], 150 mM NaCl, 0.5% NP-40, 0.5% Triton X-100, and protease inhibitor cocktail) for 30 min. The supernatants were collected via centrifugation at 13,000 rpm for 20 min at 4 °C. The protein extract was incubated with desired primary Abs, overnight at 4 °C with rotation, then coincubated with the equilibrated magnetic beads for 4 h. Beads were collected and washed three times. Then, the beads were boiled at 100 °C for 5 min in 1×SDS protein loading buffer twice and analyzed by immunoblotting.

**DARTS analysis**. Cell lysates were obtained from 3T3-L1 or HepG2 cells. Cells were scraped and lysed with M-PER lysis buffer. After centrifugation for 15 min at $16,000 \times g$, the supernatant was obtained, and protein content was quantified using Bradford reagent. Before drug treatment, the samples were diluted to achieve a protein concentration of 1 mg/mL. Samples were treated with the Kaem or DMSO for 2 h at 25 °C and then incubated with pronase (5, 10, and 20 μg/mL) or distilled water for 10 min at 25 °C. After the reaction, SDS was added to the sample and the samples were heated at 100 °C. A portion of each sample was used for LC–MS/MS analysis. Sample preparation and proteome analysis were conducted as indicated in the previous publication[69]. For western blot analysis, VDAC1 or $Na^+K^+$ ATPase was used as an internal control. For the structure–activity-relationship (SAR) analysis, kaempferol (Sigma-Aldrich, 60010), Acacetin (Sigma-Aldrich, 00017), isosakuranetin (Sigma-Aldrich, PHL82569), Biochanin A (Sigma-Aldrich, D2016), (−)Epicatechin (Sigma-Aldrich, E4018), Genistein (Sigma-Aldrich, G6649) were used.

**GCaMP3-ML1 $Ca^{2+}$ imaging**. Cells were grown on 15 mm coverslips and transfected with a plasmid encoding a perilysosomal GCaMP3-ML1 $Ca^{2+}$ probe. After 48 h, cells were stained with lysotracker and lysosomal $Ca^{2+}$ release was measured in a basal $Ca^{2+}$ solution containing 145 mM NaCl, 5 mM KCl, 3 mM MgCl2, 10 mM glucose, 1 mM EGTA, and 20 mM HEPES (pH 7.4) with or without Kaem, by monitoring fluorescence intensity at 470 nm with an LSM880 confocal microscope (Zeiss). For glycyl-L-phenylalanine-β-naphthylamide (GPN) pretreatment experiment, 48 h transfection with GCaMP3-ML1, cells were trypsinized and plated onto a glass-bottom plate. The experiment was carried out 3–5 h after plating when cells still exhibited round morphology. lysosomal $Ca^{2+}$ release was measured in a basal $Ca^{2+}$ solution containing 145 mM NaCl, 5 mM KCl, 3 mM MgCl2, 10 mM glucose, 1 mM EGTA, and 20 mM HEPES (pH 7.4) with or without GPN (100 μM) pretreatment, by monitoring fluorescence intensity at 470 nm with an LSM880 confocal microscope (Zeiss).

**Fluorescence staining**. For acridine orange (AO) stain in the screen, HeLa cells were grown in 96-well plates. Cells were treated with DMSO or indatraline, bafilomycin A1, and 658-natural chemicals for 24 h, and stained with 5 μg/mL AO. Fluorescence intensity was measured by victor plate reader, where fluorescence intensity of each well in the plate is promptly displayed as numerical readout. For confocal microscopy, 3T3-L1 or HeLa cells were grown on 15 mm coverslips at a density of $1.0 \times 10^5$ cells/well in 6-well plates The cells were then treated with drugs for the time periods indicated, followed by treatment with 5 μg/mL AO (Sigma-Aldrich). Nuclei were stained with Hoechst. Following incubation for 20 min, the cells were fixed with 4% PFA and washed three times with PBS. Images were obtained using an LSM880 confocal microscope at ×400 magnification. Red fluorescence intensity was quantified using Image J2 software. For BODIPY FL-Pepstatin A stain, HeLa cells were grown on 15 mm coverslips at a density of $1.0 \times 10^5$ cells/well in 6-well plates. The cells were then treated with drugs for the time periods indicated, followed by treatment with 1 μM BODIPY FL-Pepstatin A (Invitrogen) for 30 min. Nuclei were stained with Hoechst. Following incubation for 20 min, the cells were fixed with 4% PFA and washed three times with PBS. Images were obtained using an LSM880 confocal microscope at 400× magnification. For DQ-BSA analysis, HeLa cells were grown on 15 mm coverslips at a density of $1.0 \times 10^5$ cells/well in 6-well plates. The cells were then treated with 10 μg/mL DQ-BSA for 2 h. After change medium, cells were treated with drugs for the time periods indicated. Nuclei were stained with Hoechst with incubation for 20 min, the cells were fixed with 4% PFA and washed three times with PBS. Images were obtained using an LSM880 confocal microscope at ×400 magnification.

**LD staining**. Fully differentiated 3T3-L1 adipocytes were treated with drugs as indicated and then fixed with 4% PFA, washed with distilled water (DW), and stained with 0.3% oil-red-O solution for 1 h at room temperature. After washing

with DW three times, images were obtained using an optical microscope (Olympus IX71). After imaging, cells were dried overnight at room temperature, and then isopropanol was added. The absorbance of extracted oil-red-O dye was measured using a plate reader (Victor3, Perkin Elmer). Mouse liver tissues were sliced into sections of 10-μm thickness using a cryotome. Dried sections were fixed with 10% PFA. Fixed sections were rinsed with DW three times, plated in 100% propylene glycol, and stained with 0.5% oil-red-O for 10 min in a 60 °C oven. The sections were then washed with 85% propylene glycol and DW two times, followed by H&E staining. Images were obtained using an optical microscope. Fully differentiated 3T3-L1 adipocytes were treated with drugs as indicated, followed by treatment with BODIPY 493/503. Nuclei were stained with Hoechst. Following incubation for 20 min, samples were fixed with 4% PFA and washed with PBS three times. Images were obtained using an LSM880 confocal microscope at ×400 magnification. Green fluorescence intensity was quantified using ImageJ2 software.

**In silico docking study**. All molecular docking analyses were performed using Discovery Studio 2018 software (Accelrys, San Diego, CA, USA), adopting the CHARMm force field. The crystal structure of the bovine TUFM tr-type G domain (PDB ID 1XB2) was obtained from the RCSB protein data bank. The protein structures of bovine TUFM were energy-minimized using the Powell algorithm. The ligands were docked using Ligandfit. The Ligandfit parameters were validated using the ligand from the bovine TUFM crystal structure with 10 poses generated. The most predictive binding modes were determined based on various scoring functions, and binding energies were calculated using Ligandfit.

**Induction and treatment of high-fat diet-induced obesity mouse models**. Animal studies were approved and performed in accordance with the guidelines of the Institutional Animal Care and Use Committee of Yonsei University and conformed to the Guide for Care and Use of Laboratory Animals published by the US National Institutes of Health (The National Academies Press, 8th Edition, 2011). Four-week-old male C57BL/6j mice (Raon bio, Korea) were maintained under a 12-h light/12-h dark cycle and fed a chow diet or high-fat diet for 8 weeks. When the average bodyweight approached 40 g, the mice were randomly divided into one chow-diet group and two obese groups and treated with Kaem (10 mg/kg) or vehicle via intraperitoneal injection every 2 days for 8 weeks. During the observation period, the mice were weighed, and immediately prior to the killing, the non-fasting blood glucose level was measured. Liver and fat tissues were obtained, and tissue sections prepared as described above were subjected to optical and confocal microscopy (LSM880).

**Statistics and reproducibility**. All data are expressed as the mean ± SD or mean ± SEM, as determined using GraphPad Prism (ver. 5.00 for Windows; GraphPad Software, Inc., San Diego, CA, USA). Quantitative data were obtained from at least three independent experiments unless differently specified. Statistical analyses were performed using an unpaired, two-tailed student's t-test, with a P-value of less than 0.05 considered statistically significant (*indicates $P < 0.05$, **indicates $P < 0.01$, ***indicates $P < 0.001$).

**Reporting summary**. Further information on research design is available in the Nature Research Reporting Summary linked to this article.

## Data availability
The data generated or analyzed during this study are included in this published article and related Supplementary information files (Supplementary Data 1–8) or are available upon reasonable request. Full blots are shown in Supplementary Information.

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

## Acknowledgements
This work was partly supported by grants from the National Research Foundation of Korea, by the government of the Republic of Korea (MSIP; 2015K1A1A2028365, 2016K2A9A1A03904900, 2018M3A9C4076477), and by the Brain Korea 21 Plus Project and ICONS (Institute of Convergence Science), Yonsei University.

## Author contributions
D.K. and H.-Y.H. contributed equally. D.K., H.-Y.H., and H.J.K. participated in the conception of the project and experimental design. D.K. performed cell and molecular biology assays, in silico docking analyses, DARTS assays, and analyzed the data. D.K. and H.-Y.H. performed DIO mouse model assays. H.-Y.H., E.S.J., J.Y.K., and J.S.Y. analyzed the mass spectrometry data. D.K., H.-Y.H., and H.J.K. wrote the paper. All authors edited and approved the final manuscript.

## Competing interests
The authors declare no competing interests.
