## [Peer Review File · Communications Biology]

Reviewers' comments:

Reviewer #1 (Remarks to the Author):

This manuscript describes identification of natural product autophagy enhancer “kaempferide (Kaem) and elucidation of its mechanism. The author conducted phenotype-based screen and identified Kaem, a known simple flavonoid as autophagy enhancer. Although the activity was moderate, the author tried to elucidate the effect in vitro and vivo. Their numerous efforts are worthy of praise, however, there are many concerns in this manuscript.

Because kaempferide is very simple natural products, structure activity relationships is important. There are many commercially available related flavonoids. Please add the information of SAR.

First, they investigated the effect of Kaem on Hela cells, then moved to 3T3-L1 cells to focus on lipophagy activity. Compared to the Hela case, the decrease of p62 and LC3BII are not clear in 3T3-L1 (Fig. 3 g, h, i). Also, in Sup Fig. 12, why p62 was not decreased in control experiments?

In Fig 4f vivo experiments, LC3 seems to be increased in Kaem case. Kaem increased the expression of LC3? But, in Sup Fig7, LC3 (Atg8) was not found in increased proteins. Are there different mechanisms between in vitro and vivo?

They determined the target protein of Kaem using DARTS and LC-MS/MS analysis. They described that there are 10 protein candidates as shown in Fig 5a. Where is TUFM protein in Fig 5a? Also, there are mitochondrial related proteins in these candidates such as ATP5I (mitochondrial ATP synthase subunit) and CX7A (cytochrome C oxidase subunit). The stability of TUFM against pronase in the presence of Kaem seems to be not strong (Fig 5b and c). How about the results of ATP5I and CX7A? Because the increase effect of Kaem on calcineurin is stronger (Sup Fig 8) than the effect of p62 decrease, the ROS generation by affecting mitochondrial proteins (such as ATP5I and CX7A) seems to be main mechanism. In addition, co-IP results (Fig 6a) does not show strong interaction increase between TUFM and Atg12 compared to the amount of both input. Taken together, this reviewer feels the main mechanism of Kaem is not activation of TUFM.

This reviewer suggests re-submit revised manuscript, after the addition of re-investigation of main target of Kaem.

Reviewer #2 (Remarks to the Author):

In this manuscript, the authors identify a natural compound called kaempferide (Kaem) that induce autophagy through a mechanism involving TUFM and TRPML1/CaN/TFEB. The work is potentially interesting although there are major concerns that exclude its publication in this journal.

1- The authors identify Kaem by using an autophagy phenotype-based screen using acridine orange. The choosing of acridine orange is quite surprising since it is quite unspecific for autophagy compared with much more and widely accepted read-outs for autophagy such as LC3, p62, etc.... What is the rationale of using acridine orange?

Moreover, no data about statistical validation of this assay is presented, signal window, z-scores, positive and negative controls indicating that this assay is robust for screening and that acridine

orange is a good marker of autophagy induction...what are the criteria of hit selection?

2- The induction of TFEB nuclear translocation by an mTORC1 independent mechanism requires the presentation of mTORC1 substrate phosphorylation in the main figure.

3- In several figures, important controls are not reported. For instance, a) in Figure 2G untreated controls transfected with calcineurin plasmids; b) In Figure 2i, a control pretreating with GPN to deplete lysosomal calcium and demonstrate that Kaem is specifically inducing lysosomal calcium release.

4- From line 194, now the authors use acridine orange to monitor lysosomal activity? acridine orange is not a formal marker of lysosomal activity, maybe can be considered a marker of lysosomal acidification (acridine orange is a weak base). Other markers such as pepstatin-bodipy or magic red that bind cathepsins are better markers of lysosomal activity. To claim that Kaem enhances lysosomal function these experiments are required. How the authors exclude that Kaem is not accumulating within the lysosomes (lysosomotropic compound?)

5- Line 219, the authors made a claim by citing a reference but do not show any experimental evidence supporting it...

5- It is not clear whether the TUFM role in mtROS is deleterious for the mitochondrial function, and therefore the induction of autophagy is just a secondary effect of mitochondrial damage. If this is the case, the positive effect of LD clearance might be relatively beneficial. An analysis of mitochondrial function in vitro and in vivo is required.

minor

- The ordering of the supplementary figures is not following the results, the continuous back and forward is very confusing for the reader

- There are many typos (i.e. line 160, calciNUerin) in the text. Please make some editing

The responses to the reviewers:

We have made our best effort to address reviewers' concerns and revised the manuscript accordingly. Followings are point-to-point responses to comments by the reviewers.

- Revised portions were highlighted in red in the revised manuscript.

Reviewers' comments:

Reviewer #1:

This manuscript describes identification of natural product autophagy enhancer “kaempferide (Kaem) and elucidation of its mechanism. The author conducted phenotype-based screen and identified Kaem, a known simple flavonoid as autophagy enhancer. Although the activity was moderate, the author tried to elucidate the effect in vitro and vivo. Their numerous efforts are worthy of praise, however, there are many concerns in this manuscript.

#1. Because kaempferide is very simple natural products, structure activity relationships is important. There are many commercially available related flavonoids. Please add the information of SAR.

Response: We agree with the reviewer’s thoughtful points that additional information of SAR is needed to prove direct interaction between Kaem and TUFM. Flavonoids, to which Kaem belongs, are a structurally diverse group featuring common diphenylpropane (C6-C3-C6) core basic skeleton. The variation on the structure is discernable in the degree of oxidation of the C ring and in the substituents of the A and/or B rings. Both Kaem and kaempferol (Krol) belong to flavonol subgroup that features double bond and hydroxyl group in C ring. Based on the structural similarity between the compounds Kaem and Krol, except methoxyl and hydroxyl group on the B ring of each compound, we assessed structure-target binding activity-relationship (SAR) analysis via DARTS assay. Compared to prior result of TUFM stabilization by Kaem in a dose-dependent manner shown in Fig. 5d-f, Krol maintained TUFM stabilization with lower affinity (Saturation-EC₅₀ concentration of 74.0 μM) (Supplementary Fig. 11b-c). This result indicated that methoxyl group on B ring is required to bind to TUFM tightly. From the primary SAR information, other compounds belonging to diverse subgroups including flavone (acacetin), flavanone (isosakuranetin), flavanol ((-)-

epicatechin), and isoflavone (biochanin A, genistein) were further investigated the structure-activity relationship of subgroup structures and methoxyl group in each group. Remarkably, compounds in isoflavone subgroup did not show any binding affinity to TUFM, and methoxyl group is pivotal to bind to TUFM regardless of to which subgroups compounds belong. This SAR result provides the better information on how to regulate TUFM in a specific manner. We added the new data in Supplementary Fig. 11a-f and described this notion in the Discussion part (Page 23-24, Lines 495-513) in the revised manuscript.

Supplementary Fig. 11

Supplementary Fig. 11 Structure-activity relationship of flavonoids against TUFM. (a) Structures of flavonoid chemicals employed in this study. (b-c) 3T3-L1 cell lysate was treated with pronase for 10 minutes with or without kaempferol (Krol) pre-treatment in dose dependent manner (0, 10, 20, 50, 100, 200, and 500 μ M). Western blot analysis was subjected using indicated antibodies. Representative images (b) and sigmoidal curve of band intensity (c). (d-f) 3T3-L1 cell lysate was treated with pronase for 10 minutes with indicated chemicals pre-treatment (100 μ M). Western blot analysis was subjected using indicated antibodies. Representative images (d) and intensity of TUFM (e) and VDAC1 (f) immunoblot bands. Graph shows mean \pm SD from three independent experiments.

#2. First, they investigated the effect of Kaem on Hela cells, then moved to 3T3-L1 cells to focus on lipophagy activity. Compared to the Hela case, the decrease of p62 and LC3BII are not clear in 3T3-L1 (Fig. 3 g, h, i). Also, in Sup Fig. 12, why p62 was not decreased in control experiments?

Response: Thank you for very valuable comments. We agree with the reviewer's critical point. Since the images of p62 and LC3 of 3T3-L1 are not clear to analyze, we optimized experimental condition of immunoblotting for 3T3-L1 cell lysate, then analyzed again and replaced representative images of the result. In case of Fig. S12 (revised to Supplementary Fig. 9), the result exhibited that the protein level of p62 increased until 24 h, then decreased at 48 h. It is reasonable because many reports revealed that p62 expression is upregulated upon autophagy induction before its degradation, as a target gene of autophagy transcriptional regulators [1-

3]. Therefore, we explain that p62 exhibited fluctuating kinetics upon Kaem treatment, thus we focused the period 24-48 h that indicates p62 decrease. In siRNA control, siRab9, and siRab5 transfection group, Kaem finally promoted p62 degradation at 48 h. However, upon knockdown of Atg7, p62 degradation upon Kaem treatment was abolished in Supplementary Fig. 9a-b. In addition, the time point of p62 induction-degradation cycle seems depending on the experimental conditions such as cellular contact inhibition, vehicle treatment [4, 5]. Therefore, though the p62 protein level exhibited no difference between Kaem and DMSO vehicle control at 48 h, the cycle seemed to be affected by the experimental condition, especially vehicle material for transfection in this experiment. However, we removed control DMSO time course data in the revised manuscript to show consistency with other depicted results accordingly.

[1] Settembre C, Di Malta C, Polito VA, Garcia Arencibia M, Vetrini F, Erdin S, et al. TFEB links autophagy to lysosomal biogenesis. *Science* 2011 Jun 17;332(6036):1429-33.

[2] Pan HY, Alamri AH, Valapala M. Nutrient deprivation and lysosomal stress induce activation of TFEB in retinal pigment epithelial cells. *Cell Mol Biol Lett* 2019;24:33.

[3] Guha P, Tyagi R, Chowdhury S, Reilly L, Fu C, Xu R, et al. IPMK Mediates Activation of ULK Signaling and Transcriptional Regulation of Autophagy Linked to Liver Inflammation and Regeneration. *Cell Rep* 2019 Mar 5;26(10):2692-703 e7.

[4] Pavel M, Renna M, Park SJ, Menzies FM, Ricketts T, Fullgrabe J, et al. Contact inhibition controls cell survival and proliferation via YAP/TAZ-autophagy axis. *Nat Commun* 2018 Jul 27;9(1):2961.

[5] Song YM, Song SO, Jung YK, Kang ES, Cha BS, Lee HC, et al. Dimethyl sulfoxide reduces hepatocellular lipid accumulation through autophagy induction. *Autophagy* 2012 Jul 1;8(7):1085-97.

#3. In Fig 4f *in vivo* experiments, LC3 seems to be increased in Kaem case. Kaem increased the expression of LC3? But, in Sup Fig7, LC3 (Atg8) was not found in increased proteins. Are there different mechanisms between *in vitro* and *vivo*?

Response: Thank you for very critical comments. As the reviewer pointed out, LC3 levels appear to be different depending on the experimental scale *in vivo* or *vitro*. We firstly explain the reason why LC3 exhibited low level in the cells *in vitro*. After conversion of LC3-□ into LC3-□ upon autophagy induction to form autophagosomes, the protein itself undergoes final degradation by autophagic-turnover [6]. For that, looking at the amount of LC3 at a certain point in time is not appropriate to indicate autophagy flux. Therefore, we investigated LC3 protein level in time course manner in Supplementary Fig. 3a where the protein level

exhibited increase followed by decrease, indicating 1-cycle of autophagy acutely induced by Kaem *in vitro*. On the other hand, in animal model, we treated Kaem chronically on every 2 days for 2 months. Thus, we believe that the chemical-action had must be prolonged during the period, which could enhance and sustain autophagy with increased expression of LC3 protein level.

[6] Mizushima N, Yoshimori T. How to interpret LC3 immunoblotting. *Autophagy* 2007 Nov-Dec;3(6):542-45.

#4. They determined the target protein of Kaem using DARTS and LC-MS/MS analysis. They described that there are 10 protein candidates as shown in Fig 5a. Where is TUFM protein in Fig 5a? Also, there are mitochondrial related proteins in these candidates such as ATP5I (mitochondrial ATP synthase subunit) and CX7A (cytochrome C oxidase subunit). The stability of TUFM against pronase in the presence of Kaem seems to be not strong (Fig 5b and c). How about the results of ATP5I and CX7A? Because the increase effect of Kaem on calcineurin is stronger (Sup Fig 8) than the effect of p62 decrease, the ROS generation by affecting mitochondrial proteins (such as ATP5I and CX7A) seems to be main mechanism. In addition, co-IP results (Fig 6a) does not show strong interaction increase between TUFM and Atg12 compared to the amount of both input. Taken together, this reviewer feels the main mechanism of Kaem is not activation of TUFM.

Response: We are grateful to the reviewer for pointing out a mistake we made. EFTU is alias symbol of TUFM (protein nomenclature based on UNIPROT). We revised all of the names of proteins in the Heat-map to be depicted in their formal names (based on HGNC) (Figure 5a). In co-IP results (Figure 6a), we additionally checked β -tubulin level. Both TUFM and β -tubulin levels in the amount of input were similar among control, erlotinib, and Kaem treated groups. Atg5-Atg7 is considered to be essential molecules for the induction of autophagy. In general, the expression of Atg5-Atg12, Atg7, LC3-II is increased in the conditions of autophagy induction [7, 8]. Therefore, increased Atg12-Atg5 conjugate generation in input is expected as additional effects on Kaem's autophagy inducing activity. In addition, the functional reduction of TUFM using silencing RNA abolished Kaem-induced autophagic degradation (Figure 7d) and Kaem-mediated degradation of LDs (Figure 7f). These results demonstrate that TUFM is a biologically relevant target protein for Kaem-induced autophagy. However, we understand the reviewer's thoughtful point that other target candidates identified through DARTS LC-MS/MS analysis could be relevant targets of Kaem. In this regard, we further

investigated mitochondrial ETC proteins, cytochrome c oxidase subunit 7A2 (CX7A2, revised to COX7A2) or ATP synthase subunit e (ATP5I, revised to ATP5ME) among the target candidates accordingly. Reduction of ATP5ME expression level by assessing silencing RNA enhanced autophagy with p62 degradation 0.91-fold (Supplementary Fig. 13d), where variation of the protein expression did not affect enhancement of mtROS though (Supplementary Fig. 13g-i). COX7A2, on the other hand, enhanced autophagy with p62 degradation 0.89-fold conveying upregulation of mtROS 1.16-fold under overexpression condition, thus indicating that COX7A2 can be another possible target protein relevant with Kaem-induced autophagy. However, in comparison between TUFM and COX7A2, autophagy activity of TUFM overexpression was much likely to mimic chemical-action of Kaem, which exhibited p62 degradation 0.82-fold and 0.78-fold respectively (Fig. 7a-b), rather than COX7A2 overexpression that showed minor effect on autophagy. Although COX7A2 could partially contribute to affecting the biological activities of Kaem, consequently, these results demonstrate that TUFM is a highly relevant target protein responsible for Kaem-induced autophagy and LD degradation activities among the selected proteins of candidate. We described this notion in the Discussion part (Page 22-23, Lines 479-494) in the revised manuscript.

[7] Ye X, Zhou XJ, Zhang H. Exploring the Role of Autophagy-Related Gene 5 (ATG5) Yields Important Insights Into Autophagy in Autoimmune/Autoinflammatory Diseases. *Front Immunol* 2018;9:2334.

[8] Zheng W, Xie W, Yin D, Luo R, Liu M, Guo F. ATG5 and ATG7 induced autophagy interplays with UPR via PERK signaling. *Cell Commun Signal* 2019 May 6;17(1):42.

Supplementary Fig. 13

Supplementary Fig. 13 Kaem slightly enhances mitochondrial ROS production in 3T3-L1 cells, and other ROS relevant mitochondrial target candidates regulates mitochondrial ROS and autophagy in HeLa cells. (c-f) HeLa cells were transfected with siRNA (c,d) or clone (myc/DDK tagged) (e,f) for COX7A2 and ATP5I for 24 h. Cell extract was subjected to western blot analysis using antibodies against p62, COX7A2, ATP5I, and Myc. Representative images with the numbers for intensity of p62 immunoblot bands normalized to β -actin. (g-i) HeLa cells were transfected with siRNA (g) or clone (myc/DDK tagged) (i) for COX7A2 and ATP5I for 24 h. Cells were stained with mitoSOX and confocal microscopy performed (upper). Fluorescence intensity histogram generated using ImageJ 2 (lower). Scale bar, 50 μ m.

Reviewer #2:

In this manuscript, the authors identify a natural compound called kaempferide (Kaem) that induce autophagy through a mechanism involving TUFM and TRPML1/CaN/TFEB. The work is potentially interesting although there are major concerns that exclude its publication in this journal.

#1. The authors identify Kaem by using an autophagy phenotype-based screen using acridine orange. The choosing of acridine orange is quite surprising since it is quite unspecific for autophagy compared with much more and widely accepted read-outs for autophagy such as LC3, p62, etc.... What is the rationale of using acridine orange? Moreover, no data about statistical validation of this assay is presented, signal window, z-scores, positive and negative controls indicating that this assay is robust for screening

and that acridine orange is a good marker of autophagy induction...what are the criteria of hit selection?

Response: Thank you for very constructive comments. Discovering small molecules that enhance lysosomal functionality can be an effective strategy for targeting metabolic disorders such as obesity and diabetes, as they act as activators of autophagic-turnover [9, 10]. AO staining is a well-known assay to investigate the function and integrity of lysosome, which is also used to evaluate the status of autophagic flux [11, 12]. Our goal is to find out new small molecules that enhance autophagic-turnover, which has been focused activities targeting metabolic diseases like obesity and diabetes, within the cells. Therefore, we focused on staining methods indicating an increase of acidic lysosomes as one of hallmarks of autophagy. Moreover, AO is a weakly basic dye that easily penetrates cell membranes and is retained in the cellular compartments with low pH, resulting in bright orange fluorescence of lysosomes in live cells [13]. Many papers studying autophagy reported their use of AO to indicate lysosomal acidic status [14-16]. We thus leveraged this method to our screen system. In respect with reviewer's concern about choosing acridine orange (AO) instead of other widely accepted read-outs in autophagy activity, we added the rationale of choosing this screening method (Page 6, Line 107-111). We agree with the reviewer's comment that the screening is a bit robust (since n=1). Therefore, we tried to tightly validate autophagy activity of the hit to make up for the screening result as shown in Figure 1-3. In the screening assess, we additionally checked that a positive control indatraline (Inda), which enhances lysosomal acidity [17], increased AO intensity 1.2-fold, and negative control bafilomycin A1, which inhibits acidic lysosome by perturbing proton channel [15], decreased AO intensity 0.7-fold. These results were added in the revised manuscript (Supplementary Fig. 1a, c-d). As we simply described that TOP-2 compounds (Kaem and tiliroside) were selected as hits (over1.4-fold) without logical description in the previous submitted manuscript, the hit selection process is additionally provided in the revised manuscript. Among 13 candidates that enhanced AO intensity over then 1.2-fold (standard based on positive control, Inda), we first excluded some compounds that were previously reported as autophagy regulators to identify a NEW autophagy enhancer per purpose of this study. Then we checked whether the other hit compounds meet constraints of Lipinski's rule of five (MW, LogP, hydrogen donor and acceptor, and rotational bond), and then checked autophagy protein markers in the cells. Kaem was selected as a final hit covering all of these constraints (Page 6, Line 112-124).

Albeit the screening is a bit robust (since n=1), we suggested Kaem as a new autophagy enhancer through tight validation of autophagy activity leveraging diverse assays (Page 7-8, Line 125-161).

- [9] Rocchi A, He C. Emerging roles of autophagy in metabolism and metabolic disorders. *Front Biol (Beijing)* 2015 Apr;10(2):154-64.
- [10] Lim H, Lim YM, Kim KH, Jeon YE, Park K, Kim J, et al. A novel autophagy enhancer as a therapeutic agent against metabolic syndrome and diabetes. *Nat Commun* 2018 Apr 12;9(1):1438.
- [11] SenthilKumar G, Skiba JH, Kimple RJ. High-throughput quantitative detection of basal autophagy and autophagic flux using image cytometry. *Biotechniques* 2019 Aug;67(2):70-73.
- [12] Hwang HY, Cho YS, Kim JY, Yun KN, Yoo JS, Lee E, et al. Autophagic Inhibition via Lysosomal Integrity Dysfunction Leads to Antitumor Activity in Glioma Treatment. *Cancers (Basel)* 2020 Feb 27;12(3).
- [13] Byvaltsev VA, Bardanova LA, Onaka NR, Polkin RA, Ochkal SV, Shepelev VV, et al. Acridine Orange: A Review of Novel Applications for Surgical Cancer Imaging and Therapy. *Front Oncol* 2019;9:925.
- [14] Yue W, Hamai A, Tonelli G, Bauvy C, Nicolas V, Tharinger H, et al. Inhibition of the autophagic flux by salinomycin in breast cancer stem-like/progenitor cells interferes with their maintenance. *Autophagy* 2013 May;9(5):714-29.
- [15] Li X, Zhu F, Jiang J, Sun C, Zhong Q, Shen M, et al. Simultaneous inhibition of the ubiquitin-proteasome system and autophagy enhances apoptosis induced by ER stress aggravators in human pancreatic cancer cells. *Autophagy* 2016 Sep;12(9):1521-37.
- [16] Pahari S, Negi S, Aqdas M, Arnett E, Schlesinger LS, Agrewala JN. Induction of autophagy through CLEC4E in combination with TLR4: an innovative strategy to restrict the survival of *Mycobacterium tuberculosis*. *Autophagy* 2020 Jun;16(6):1021-43.
- [17] Cho YS, Yen CN, Shim JS, Kang DH, Kang SW, Liu JO, et al. Antidepressant indatraline induces autophagy and inhibits restenosis via suppression of mTOR/S6 kinase signaling pathway. *Sci Rep* 2016 Oct 3;6:34655.

#2. The induction of TFEB nuclear translocation by an MTORC1 independent mechanism requires the presentation of mTORC1 substrate phosphorylation in the main figure.

Response: We appreciate this Reviewer's valuable suggestion. We examined canonical substrate of MTORC1, P70S6K to investigate the upstream molecular cascade associated with Kaem-induced TFEB activation. Likely to mTOR, its substrate P70S6K remained in the phosphorylated state (T389) from the early (0.5 h) to late (24 h) phases of the experiment following Kaem treatment, although the

known mTOR inhibitor rapamycin significantly inhibited phosphorylation over this time course. We added the new data in Figure 2d-f and described the notion in the main text (Page 9, Line 177-182).

Fig. 2

Fig. 2 Kaem induces TFEB translocation to the nucleus via Ca^{2+} signaling regulation without mTOR inhibition. (d-f) HeLa cells were treated with DMSO control, rapamycin (Rapa), and Kaem respectively for indicated period. Cell extracts were subjected to western blot analysis using antibodies against p-mTOR, mTOR, p-P70S6K, and P70S6K. Representative images (d), intensity of p-mTOR immunoblot bands normalized to mTOR (e), intensity of p-P70S6K immunoblot bands normalized to P70S6K (f). Kaem, 20 μ M; Rapa, 10 μ M. Graph shows mean \pm SD from three independent experiments.

#3. In several figures, important controls are not reported. For instance, a) in Figure 2G untreated controls transfected with calcineurin plasmids; b) In Figure 2i, a control pretreating with GPN to deplete lysosomal calcium and demonstrate that Kaem is specifically inducing lysosomal calcium release.

Response: Thank you for very constructive comments. We added quantitation of untreated control (DMSO treated) in Figure 2i accordingly. To validate that Kaem specifically induces lysosomal calcium release, lysosomotropic compound glycyl-L-phenylalanine- β -naphthylamide (GPN) was pretreated. The pretreatment of GPN abolished Kaem induced responses of GCaMP3-ML1 fluorescence (Supplementary Fig. 4b). We added the new data in Figure 2i and Supplementary Fig. 4b respectively, and described the notion in the main text (Page 10, Line 205-207).

Supplementary Fig. 4

Supplementary Fig. 4 Kaem induces autophagy through lysosomal Ca^{2+} -TFEB regulation without mTOR perturbation. (b) HeLa cells were transfected with GCaMP3-ML1 encoding a lysosome-specific Ca^{2+} probe and then treated with DMSO control (Basal) or Kaem, with or without glycy-L-phenylalanine- β -naphthylamide (GPN) pretreatment for 1 h. Lysosomal Ca^{2+} release was visualized by confocal microscopy. Kaem, 20 μM , GPN, 100 μM , Scale bar, 20 μm .

#4. From line 194, now the authors use acridine orange to monitor lysosomal activity? acridine orange is not a formal marker of lysosomal activity, maybe can be considered a marker of lysosomal acidification (acridine orange is a weak base). Other markers such as pepstatin-bodipy or magic red that bind cathepsins are better markers of lysosomal activity. To claim that Kaem enhances lysosomal function these experiments are required. How the authors exclude that Kaem is not accumulating within the lysosomes (lysosomotropic compound?)

Response: As be addressed with the comment #1 above, acridine orange staining is a well-known assay to examine the function and integrity of lysosome, which is also used to evaluate the status of autophagic flux. We understand the reviewer's valuable points that acridine staining is not a formal marker of lysosomal activity. For further validation of an active state of lysosome to lead autophagy by Kaem, other fluorescent probes such as BODIPY FL-pepstatin A, double quenched BSA (DQ-BSA), and lysotracker were leveraged accordingly. Since pepstatin A is a direct inhibitor of aspartic proteinases such as pepsin, cathepsins D and E, probe-stained puncta indicate active cathepsins in the lysosomes. Fluorescence analysis exhibited that BODIPY FL-pepstatin A labeling within active cathepsin-positive vacuoles was markedly increased by rapamycin and Kaem treatment, by contrast it was almost completely abolished by bafilomycin A1 treatment (Supplementary Fig. 3b). In DQ-BSA assess, which is self-quenched fluorogenic substrate that requires undergoing proteolytic cleavage in acidic compartments such as autophagy-associated cargo to be observed in fluorescence, Kaem treated cells

indicated increased number and intensity of fluorescent vacuoles where lysosomal proteolysis enhanced turn-over of autophagy-cargo (Supplementary Fig. 3c). In lysotracker analysis, Kaem increased lysosomal acidic puncta, whereas lysosomotropic agent NH₄Cl diminished all acidic vacuoles (Supplementary Fig. 3d). We added the new data in Supplementary Fig. 3b-d and described the notion in the main text (Page 7, Line 129-144).

#5. Line 219, the authors made a claim by citing a reference but do not show any experimental evidence supporting it...

Response: We agree with reviewer's valuable point that this sentence is a predictive claim that do not show any experimental data or evidence. We deleted this sentence, "due to self-enhancement of expression, as reported previously", in revised manuscript (Page 12, Line 245-246).

#6. It is not clear whether the TUFM role in mtROS is deleterious for the mitochondrial function, and therefore the induction of autophagy is just a secondary effect of mitochondrial damage. If this is the case, the positive effect of LD clearance might be relatively beneficial. An analysis of mitochondrial function in vitro and in vivo is required.

Response: Thank you for very constructive comments. It is reported that mtROS-induced oxidative stress can induce depolarization of mitochondrial membrane potential [18], thus we examined mitochondrial membrane potential by JC-1 stain. Kaem slightly reduced mitochondrial potential dose-dependently in HeLa cells, while exhibited fluctuations in 3T3-L1 adipocytes, without any inhibition of cell viability (Supplementary Fig. 14). These results indicated that Kaem somehow regulates mitochondrial function with ROS regulation in a safe way with maintaining cellular viability. Although it is true that mitochondrial damage by mtROS can induce autophagy such as mitophagy as reviewer pointed out, on the other hand, mtROS itself induces autophagy through acting as a secondary messenger in molecular cross-talk [19]. Therefore, we speculate that mtROS generated by TUFM modulation could enhance autophagy rather deleteriously affecting to the cells.

[18] Park J, Lee J, Choi C. Mitochondrial network determines intracellular ROS dynamics and sensitivity to oxidative stress through switching inter-mitochondrial messengers. *PLoS One* 2011;6(8):e23211.

[19] Roca-Agujetas V, de Dios C, Leston L, Mari M, Morales A, Colell A. Recent Insights into the

Mitochondrial Role in Autophagy and Its Regulation by Oxidative Stress. *Oxid Med Cell Longev* 2019;2019:3809308..

Minor concerns.

- The ordering of the supplementary figures is not following the results, the continuous back and forward is very confusing for the reader

Response: Thank you for pointing out these errors. The whole supplementary figures were re-aligned in serial order throughout the revised manuscript.

- There are many typos (i.e. line 160, calciNUerin) in the text. Please make some editing

Response: Thank you for pointing out these errors. The typos including the proper nouns and their abbreviations are corrected throughout the revised manuscript accordingly.

In addition to the revisions commented in the responses, minor typos and errors in statements were corrected throughout the manuscript.

Thank you.

REVIEWERS' COMMENTS:

Reviewer #1 (Remarks to the Author):

The authors fully addressed my concerns. Authors tried additional experiments and new data was also suitably added. I think it was hard to work in this COVID-19 situations. I recommend this manuscript is suitable to be accepted to this journal.

Reviewer #2 (Remarks to the Author):

I appreciate the effort made by the authors to answer all my comments/concerns. Now these concerns are addressed.